# Rapid Bayesian learning in the mammalian olfactory system

Naoki Hiratani [1]✉ & Peter E. Latham [1]

Many experimental studies suggest that animals can rapidly learn to identify odors and predict the rewards associated with them. However, the underlying plasticity mechanism remains elusive. In particular, it is not clear how olfactory circuits achieve rapid, data efficient learning with local synaptic plasticity. Here, we formulate olfactory learning as a Bayesian optimization process, then map the learning rules into a computational model of the mammalian olfactory circuit. The model is capable of odor identification from a small number of observations, while reproducing cellular plasticity commonly observed during development. We extend the framework to reward-based learning, and show that the circuit is able to rapidly learn odor-reward association with a plausible neural architecture. These results deepen our theoretical understanding of unsupervised learning in the mammalian brain.

[1] Gatsby Computational Neuroscience Unit, University College London, 25 Howland Street, London W1T 4JG, UK. ✉email: N.Hiratani@gmail.com

I t is crucial for animals to infer the identity of odors, in situations ranging from foraging to mating[1]. While some odors are hardwired[2], most must be learned. Learning, however, is particularly difficult, especially in natural environments where odors are rarely presented in isolation, most odors are presented a small number of times, and odor identities are rarely supervised. Nevertheless, animals can learn to associate an odor with a reward in a few trials[3–5]. Our goal here is to elucidate the local plasticity mechanisms that orchestrate this rapid learning.

To gain a conceptual understand of how learning occurs, note that if the affinities of olfactory receptor neurons (OSNs) to odors were known, approximate Bayesian inference could be used to infer which odors are present given OSN activity[6]. And in a supervised setting—a setting in which the animal is told which odors are present—the affinities (i.e. the weights) could be learned efficiently using recently proposed Bayesian approaches[7,8]. Here we show that, even when the weights are not known and learning is unsupervised, we can combine these two methods to simultaneously learn the weights and infer the odors.

Our approach is as follows: when inferring which odors are present, average over the uncertainty in the weights; then use the inferred odors to update the estimates of the weights, and, importantly, decrease the uncertainty. As the estimates of the weights become more accurate, inference also improves. However, while straightforward, exact implementation of this learning process is intractable. Consequently, we have to use an approximate method[9].

Although inference is approximate, our model still leads to faster learning of olfactory stimuli compared to previously proposed sparse-coding-based approaches[10–12]. It also provides some insight into olfactory circuitry: it reveals the advantage, relative to the rectified linear transfer function[13], of sigmoidal-shaped $f$–$I$ curves typical of biological neurons[14,15], and it reproduces the reduction in neuronal input gain[16,17] and learning rate[18] commonly observed during development. In addition, it predicts that the learning rate of granule cells should decrease as they become more selective, and thus exhibit lower lifetime sparseness[19,20], something that is possible (although difficult) to test experimentally. And finally, we extended our model to an odor–reward association task, and found that learning of a concentration invariant representation at the piriform cortex helps rapid odor–reward association.

While our approach gives us a model that is reasonably consistent with mammalian olfactory circuitry, the architecture predicted by our approximate Bayesian algorithm does not perfectly match the architecture of the olfactory system. However, a plausible olfactory circuit based on our model, but with the addition of recurrent inhibition among piriform neurons[21], still learns to perform reward-based learning quickly. These results suggest that even at the circuit level, approximate Bayesian optimization may underlie rapid biological learning. But at the same time, our study reveals its limitation when applied to a complicated system.

## Results

**Problem setting**. Let us denote odor concentrations by a vector $\mathbf{c}$ = $(c_1, \ldots, c_M)$, where $c_j > 0$ if odor $j$ is present and $c_j = 0$ otherwise. By odor, we mean something like the odor of apple or coffee, not a single odorant molecule. In a typical environment, odors are very sparse, in the sense that few of them have a significant presence (i.e. $c_j > 0$ for a small number of $j$ at any time; Fig. 1 left).

In the olfactory system, odors are first detected by OSNs, and then transmitted to glomeruli as spiking activity[22]. Neural activity accumulated at a glomerulus, denoted $x_i$ for $i$th glomerulus (and

thus $i$th OSN receptor type), is, approximately

$$x_i = \sum_j w_{ij} c_j + n, \qquad (1)$$

where $n$ is the noise due to sensory variability and unreliable OSN-spiking activity, and the affinity, or the mixing weight, $w_{ij}$, determines how strongly odor $j$ activates glomerulus $i$ (Fig. 1 right). OSN activity shows a roughly logarithmic dependence on odor concentration[23,24]. Thus the amplitude, $c_j$, of each odor reflects log-concentration, not concentration. Below a threshold, here taken to be zero, odors are considered undetectable.

**Olfactory learning as Bayesian inference**. The goal of the early olfactory system is to infer which odors are present and what their concentrations are, based on OSN activity, $\mathbf{x}$. However, this is a difficult problem because the animal does not know the mixing weights, $\mathbf{w}$, but instead has to learn them, without supervision. One common approach to this type of unsupervised learning is the sparse coding model. Its associated learning algorithm is, however, inefficient, and thus slow, as we will see below (see the subsection "Sparse coding" in the Methods section). We thus turn to Bayesian inference.

The Bayesian approach is efficient because it takes into account uncertainty in both odor, $\mathbf{c}$, and weight, $\mathbf{w}$, and it can naturally incorporate a prior that reflects the sparseness of the olfactory environment. The steps are straightforward: first write down, from Eq. (1), an expression for $p(\mathbf{c}|\mathbf{x}, \mathbf{w})$, the distribution over odor concentrations given glomeruli activity, $\mathbf{x}$, and weights $\mathbf{w}$; then marginalize over the distribution of the weights given all the previous inputs, $p(\mathbf{w}|$ past observations of $\mathbf{x}$) (see Methods section, Eq. (10)). However, exact marginalization is neither computationally tractable nor biologically plausible. We therefore employ a variational Bayesian approximation[9], by replacing the true joint probability distribution with a fully factorized one. The effect of making a variational approximation is illustrated in Fig. 2c: the distribution of a pair of odors are typically slightly anti-correlated (Fig. 2c, left), while the variational distribution is independent (Fig. 2c, right). Because the anti-correlation is typically weak, the variational distribution captures the true distribution well.

The derivation of the algorithm for variational inference is described in detail in Methods section; here we simply give the results. The variational probability distribution of the concentration of odor $j$ is updated iteratively as (see Methods section, Eq. (14b))

$$q(c_j|\mathbf{x}) \propto q(\mathbf{x}|c_j)p_c(c_j) \qquad (2)$$

where $q(\mathbf{x}|c_j)$ is the variational likelihood of the concentration of the $j$th odor, $c_j$, given $\mathbf{x}$, and $p_c(c_j)$ is the prior distribution over $c_j$. We take the noise, $n$, in Eq. (1) to be Gaussian, so $q(\mathbf{x}|c_j)$ is Gaussian (Fig. 2a, left). And to reflect the sparsity, $p_c(c_j)$ is taken to be a point mass at zero combined with a continuous piece at positive concentration (Fig. 2a, middle). Because, the prior strongly favors the absence of odors, the estimated mean concentration, $\langle c \rangle_{q(c|\mathbf{x})}$ (dashed black line in Fig. 2a, right), is typically smaller than the mean over the likelihood function, $\langle c \rangle_{q(\mathbf{x}|c)}$ (dashed orange line in Fig. 2a, right).

Similarly, the update rule for the variational probability distribution of a weight is given by (see Methods section, Eq. (14a))

$$q_t(w_{ij}) \propto \Delta q_t(w_{ij}, \mathbf{x})q_{t-1}(w_{ij}), \qquad (3)$$

where $\Delta q_t(w_{ij}, \mathbf{x})$ is the evidence provided by the new information, carried in $\mathbf{x}$, at trial $t$ (Fig. 2b) and $q_t(w_{ij})$ is the variational probability distribution of the weight, $w_{ij}$, given observations up to trial $t$ (we suppress the time dependence to

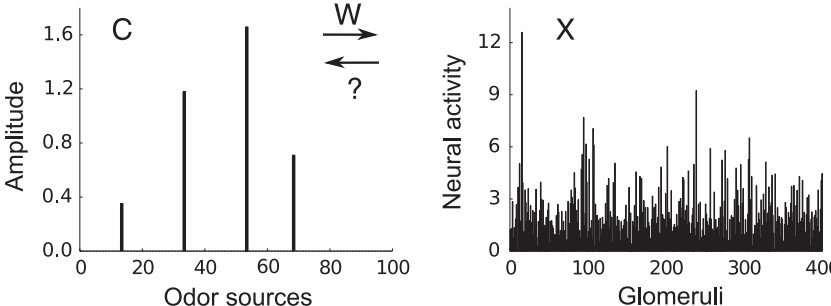

**Fig. 1 Problem setting.** An example odor stimulus, **c** (left), and the response at the glomeruli, **x** (right). The mixing weights (i.e., affinities), **w** (which are unknown to the animal) map odors, with concentration **c**, to OSN activity accumulated at the glomeruli, **x**. A goal of the animal is to infer the odor concentrations from the glomeruli activity.

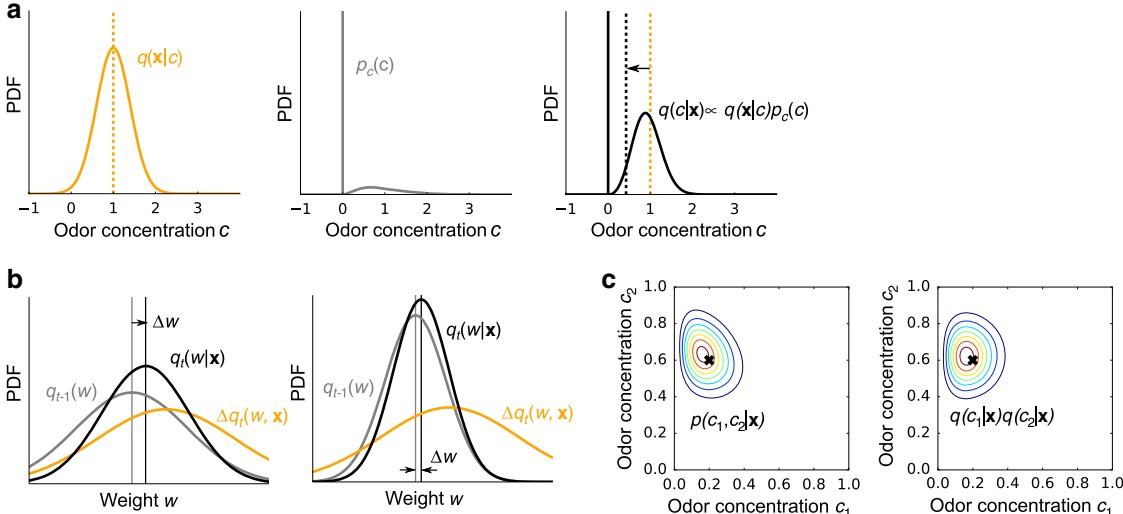

**Fig. 2 Bayesian inference of odors and weights. a** Inference of odor concentration. Combining the likelihood $q(\mathbf{x}|c)$ (left) and the prior $p_c(c)$ (middle), the posterior distribution $q(c|\mathbf{x})$ is obtained (right). The orange dashed line is the mean concentration associated with the likelihood, $q(\mathbf{x}|c)$; the black dashed line is the mean associated with the posterior, $q(c|\mathbf{x})$. Because the prior strongly favors the absence of odors, the latter is shifted to lower concentration. **b** Illustration of the weight update given the same sensory evidence $\Delta q_t(w, \mathbf{x})$ when the previously estimated probability distribution over the weights, $q_{t-1}(w)$, is broad (left), and narrow (right). Note that the mean of $q_{t-1}(w)$ is the same in both panels. **c** Illustration of the variational approximation. The true posterior over the joint distribution of odors $c_1$ and $c_2$, $p(c_1, c_2|\mathbf{x})$ (left), is approximated by a factorized distribution $q(c_1|\mathbf{x})q(c_2|\mathbf{x})$ (right). The black cross indicates the true concentrations, and colored lines are contours of equal probability.

reduce clutter). Importantly, depending on the uncertainty in the weights, the same stimulus causes different amounts of plasticity. In particular, the higher the uncertainty in the estimated weight, $w_{ij}$, at $t-1$, the larger the change in the mean weight, $\Delta w$ (left vs. right in Fig. 2b).

The update rules given in Eqs. (2) and (3) can be mapped onto neural dynamics and synaptic plasticity that closely mirrors the mammalian olfactory bulb (Fig. 3a and b). The firing rate dynamics obeys

$$\tau_r \frac{dm_i}{d\tau} = -m_i - \sum_{j=1}^{M} w_{ij}^L \bar{c}_j + x_i \qquad (4a)$$

$$\tau_r \frac{d\bar{c}_j}{d\tau} = -\bar{c}_j + F_j\left(\sum_{i=1}^{N} w_{ji}^F m_i\right) \qquad (4b)$$

where $\tau$ denotes time within an odor presentation (not to be confused with $t$, which refers to trial), $m_i$ is the firing rate of the $i$th M/T (mitral/tufted) cell relative to baseline, and $\bar{c}_j$ is the firing rate of the $j$th granule cell. The $i$th M/T cell is linearly modulated by excitatory input from glomerulus $i$, via $x_i$, and also by

inhibitory input from granule cells, the $\bar{c}_j$. The granule cells, whose activity correspond to the expected concentration of the odors, are driven by excitatory input from M/T cells, mediated by a nonlinear transfer function $F_j$. As we discuss below, this nonlinearity plays a critical role in rapid learning.

The weights in Eq. (4), $w_{ij}^F$ and $w_{ij}^L$, correspond to M/T-to-granule and granule-to-M/T synapses, respectively (blue and red arrows in Fig. 3b). These synapses jointly form a dendro-dendritic connection between M/T and granule cells[25]. To keep track of the variational probabilistic distribution $q_t(w_{ij})$, both the mean and the variance of each weight need to be updated. The update of the mean is

$$w_{ji}^{F,t} = (1 - \delta_j^{w,t})w_{ji}^{F,t-1} + \frac{1/t}{\rho_j^t \sigma_x^2}\bar{c}_j m_i \qquad (5a)$$

$$w_{ij}^{L,t} = (1 - \delta_j^{w,t})w_{ij}^{L,t-1} + \frac{1/t}{\rho_j^t \sigma_x^2}m_i \bar{c}_j \qquad (5b)$$

where $m_i$ and $\bar{c}_j$ are evaluated at the end of the odor presentation. Here $\delta_j^{w,t}$ is the discount factor and $\rho_j^t$ represents the precision

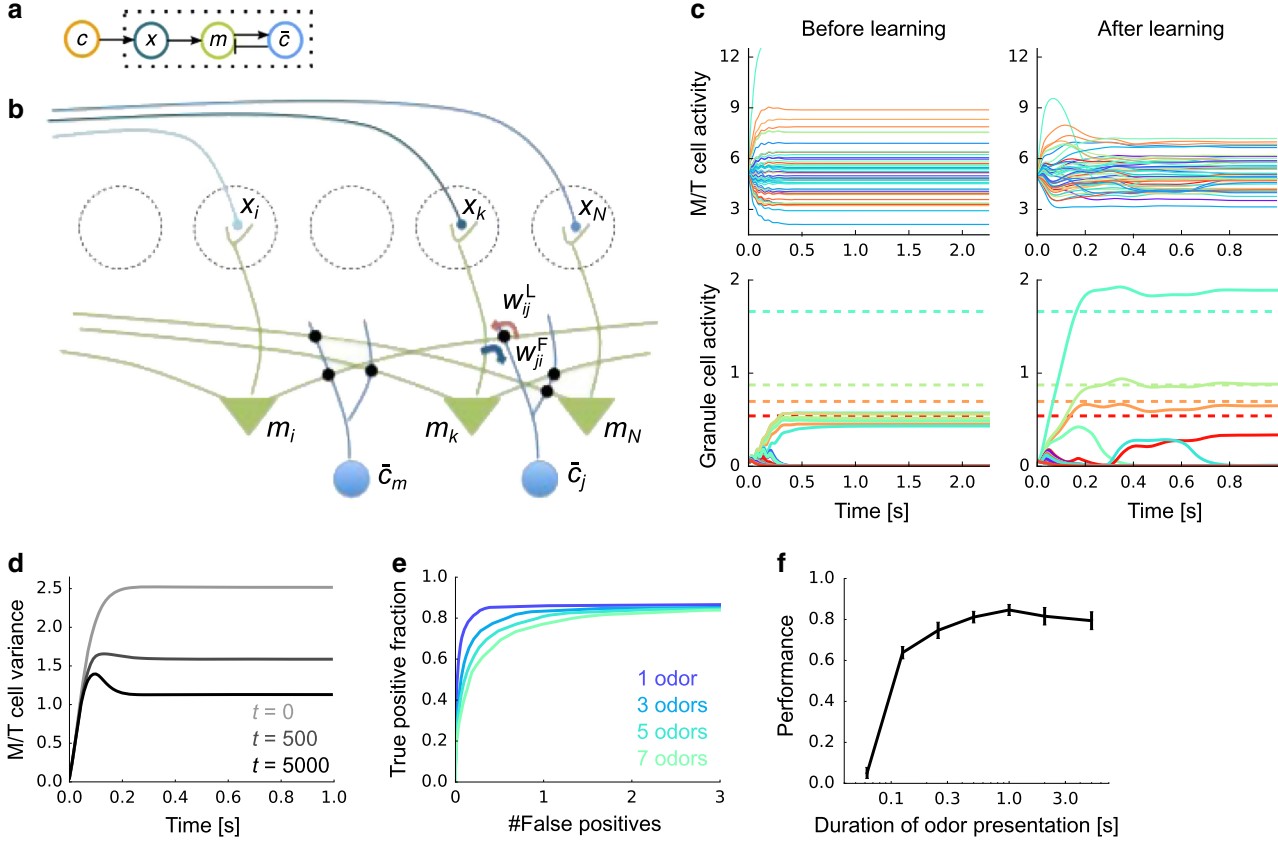

**Fig. 3 Neural implementation of Bayesian learning. a** Schematic of the neural architecture. Dotted box represents the internal variables of the brain; the odor, **c**, comes from the outside world. **b** The neural implementation of our Bayesian learning model maps almost perfectly onto the circuitry of the olfactory bulb. Dotted circles are glomeruli, green triangles are M/T cells, and blue circles represent olfactory granule cells. Red and blue arrows indicate weights from granule to M/T and M/T to granule cells, respectively. **c** An example of firing rate dynamics before (left) and after (right) learning ($M = 50$ odors, $N = 400$ glomeruli, four odors presented). Different colors represent different neurons. Dotted horizontal lines in the bottom figures represent the true concentration of the presented odors. **d** Change in the variance of M/T cells during learning ($t$: trial). The expectation was taken over both population and trials. **e** Receiver operating characteristic (ROC) curves under different numbers of simultaneously presented odors ($M = 100$ odors, $N = 400$ glomeruli). See subsection "ROC curve" in the Methods section for details. **f** Performance under learning from various odor exposure duration (see subsection "Performance evaluation" in the Methods section), where $M = 100$, $N = 400$, and three odors are presented simultaneously, on average. The lines and their error bars are mean and standard deviation over 10 simulations.

(the inverse of the variance) of the synaptic weights $w_{ji}^{F,t}$ and $w_{ij}^{L,t}$ (see subsection "Synaptic plasticity" in the Methods section for details). This rule is Hebbian, as the update depends on the product of presynaptic and postsynaptic activity $m_i$ and $\bar{c}_j$. It is also adaptive, as the update depends on the precision, $\rho_j^t$: because of the $1/\rho_j^t$ dependence, low precision (and thus high uncertainty) produces large weight changes while high precision (and thus low uncertainty) produces small weight changes. This is illustrated in Fig. 2b. The precision, $\rho_j^t$, is also updated in an activity-dependent manner (see the Methods section, Eq. (35)). Figure 3c describes typical neural dynamics before and after learning. Before learning, when a mix of four odors is presented, M/T activity quickly converges to constant values with a relatively broad range (Fig. 3c, top-left), and granule cell activity is small and homogeneous (Fig. 3c, bottom-left). After learning, M/T cells exhibit transient activity, followed by convergence to a somewhat smaller range than before learning (Fig. 3c, top-right), as the large input-driven activity is partially canceled by the feedback from the granule cells. Granule cells, on the other hand, show very selective responses, with activity levels roughly matching the concentration of the corresponding odors (Fig. 3c, bottom-right).

The activity profiles of cells in our model have many similarities with experimental observations. For instance, as observed in experiments[26], M/T cells show both positive and negative responses relative to baseline (Fig. 3c top, here the baseline is 5), and their responses become more transient after learning (Fig. 3c, top-right, and Fig. 3d). Moreover, the response range of M/T cells becomes smaller as the animal learns the odors (Fig. 3d), as observed experimentally[27]. In addition, after learning, granule cell activity is strongly modulated by odor concentration (Fig. 3c bottom-right; dotted horizontal lines represent the true concentrations of the corresponding odors), as observed experimentally[28].

After learning, the circuit can robustly detect odors with very few false positives, even when several odors are presented simultaneously (Fig. 3e). Moreover, the learning performance was robust with respect to odor presentation time: even if the odors were presented for only a few hundred milliseconds, which corresponds roughly to one sniff cycle[29,30], performance remained high (Fig. 3f). Learning was also robust to changes in the prior: a large increase in the range of possible odor concentrations had very little effect on learning performance (Supplementary Fig. 1).

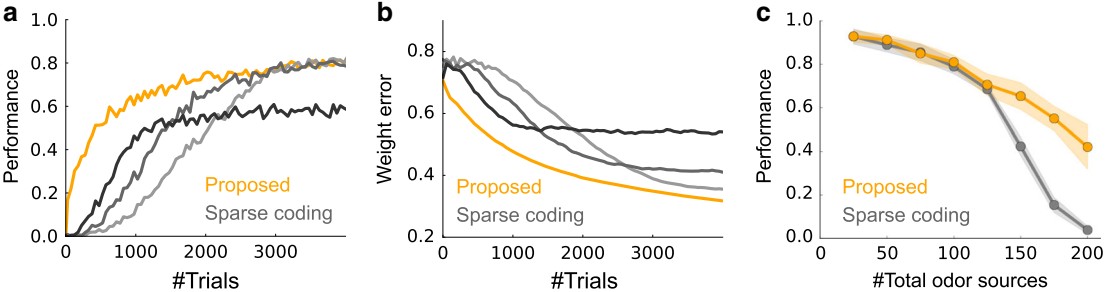

**Fig. 4 Performance comparison. a** Learning curves for our model (orange) and sparse coding (light gray to black). $M = 100$ odors, $N = 400$ glomeruli, and on average, three odors were presented at each trial. See subsection "Performance evaluation" in the Methods section for details. The learning rates of the sparse coding model, $\eta_w$, were 0.3, 0.5, and 1.0 from light gray to black. **b** Same as **a**, but the performance was evaluated by the error in the weight. **c** Performance (after learning from 4000 trials) of the proposed Bayesian model (orange) and the sparse coding model (gray) versus the number of odors. Shaded regions represent standard deviation over 10 simulations. As in panels **a** and **b**, $N = 400$ glomeruli and three odors were presented on average. Here, $\eta_w$ was fixed at 0.5.

The Bayesian approach is optimal if implemented exactly, but in the approximate model used here, learning is necessarily suboptimal. To determine how suboptimal, we would need to compare against exact inference. However, that is not feasible because exact inference is intractable. Our model does, however, do better than the sparse coding model (Fig. 4): it learns much faster (Fig. 4a), and it achieves high performance without fine tuning, whereas the learning rate of the sparse coding model must be fine-tuned (gray lines in Fig. 4a). This advantage was replicated when we assessed the performance by the error in the weights (Fig. 4b). Despite faster learning, the asymptotic performance of the Bayesian model is similar to that of sparse coding when there are a relatively small number of odor sources in the environment, and much better when there are many sources, although the performance of both models deteriorates in that regime (Fig. 4c).

These results indicate that a variational approximation of Bayesian learning and inference enables data efficient learning, and does so using biologically plausible learning rules and neural dynamics. How does our model manage to perform fast and robust learning? And is there evidence that the brain uses this strategy? Below, we show that our proposed circuit performs well because it exploits the sparseness of the odors and utilizes the uncertainty in both the weights and odor concentration. We then discuss the relationship of our model to experimental observations.

**The sparse prior leads to a nonlinear transfer function.** An important feature of olfaction, like many real world inference problems, is that the distribution over odors has a mix of discrete and continuous components: an odor may or may not be present (the discrete part), and if it is present its concentration can take on a range of values (the continuous part). In our model, we formalize this with a spike and slab prior (Fig. 2a middle): the spike is the delta function at zero; the slab is the continuous part. In this model, sparseness is ensured by setting the cumulative probability of the slab, denoted $c_o$, close to zero.

To see how the prior affects the dynamics, note that the granule cells ($\bar{c}_j$ in Eq. (4)) represent the expected concentration of the odors, and so take the prior into account. Thus, after learning, most of them have near zero activity, with only a few of them active (Fig. 3c, bottom right panel). To achieve sparsity, the granule cells need a great deal of evidence to report non-negligible concentrations. That is reflected in the transfer functions of the granule cells (the function $F_j$ in Eq. (4b); see orange curve in Fig. 5a). The function exhibits near zero response (corresponding to near zero concentration) for small input, followed by a sharp rise and then an approximately linear response for large input.

If we derive update rules using a different prior, the transfer function changes. If we then perform inference and learning using the transfer function derived under a different prior, but drawing odors from the true prior, performance is, not surprisingly, sub-optimal (see subsection "Models with various priors on odor concentration" in the Methods section). For example, if we constrain the odors to be non-negative, the transfer functions are approximately rectified linear, a commonly used nonlinearity in artificial neural networks (gray line Fig. 5a[13]). However, this model failed to learn the input structure generated from the spike-and-slab prior, as the sparseness of the odor concentration is not taken into account (gray line in Fig. 5b). If we constrain the odors to be non-negative, but also ensure that they are not too large, by introducing an exponential decay[10], learning improves initially, but the weight error eventually increases (black lines in Fig. 5a and b). These results suggest that the classic input–output function—sigmoidal at small input and linear at large input—found both in vitro[14,31] and in biophysically realistic models of neurons[15], reflects the fact that the world is truly sparse—something not captured by classical sparse coding models. These gain functions thus offer a normative explanation for the biophysical responses of typical olfaction neurons to input. The shape of the activation function for the precision update also depends on the choice of prior, but they all closely resemble the squared transfer function, $F^2$ (Supplementary Fig. 2).

As the animal learns a better approximation to the true weights, the olfactory system can extract more information from the OSN activity; this results in a change in the transfer function. In particular, the transfer function exhibits a decrease in gain with learning (mainly a shift to the right), as shown in Fig. 5c (see subsection "The variational weight distribution" in the Methods section for details). Such a decrease in gain is a widely observed phenomenon among diverse neurons during development[14,16]. It is also consistent with the reduction of input resistance observed in adult-born granule cells during development[17,18], as low resistance causes low excitability. If the transfer functions were held fixed during learning, performance would deteriorate gradually (gray and black curves vs. orange line in Fig. 5d), though the benefit of the adaptive gain was rather small in our model setting.

**Weight uncertainty leads to adaptive synaptic plasticity.** A key aspect of our model is that it explicitly takes the uncertainty of the weights into account. This leads to an adaptive learning rate (see Eq. (5)). In particular, the learning rate is the product of two terms: $(1/t) \times 1/\rho_j^t$. The first term, $1/t$, is a global decay, and

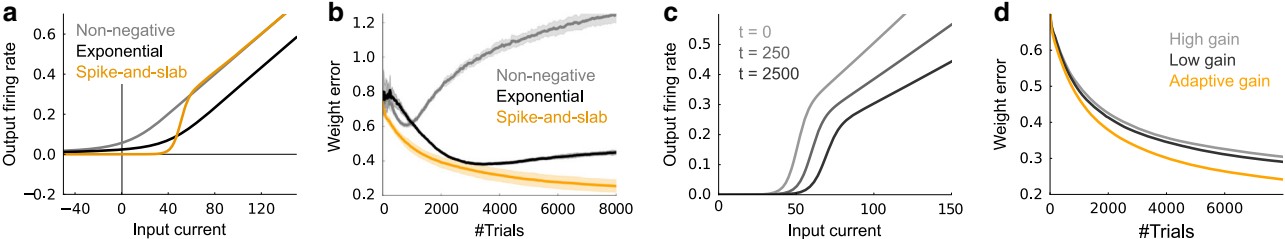

**Fig. 5 Adaptive transfer functions. a** The shapes of the transfer functions of granule cells under different priors on the odor distribution. See subsection "Models with various priors on odor concentration" in the Methods section for the details. **b** Weight errors under different priors. Shaded regions represent standard deviation over 10 simulations. **c** The average transfer function $F[y, \bar{c} = 0]$ at the beginning (light gray), middle (gray), and the end (black) of the learning. The x-axis represents the input current $y$. **d** The weight error under fixed input gain, compared to the control model with adaptive gain, averaged over 50 simulations. For the gray line, the transfer function was set to the top curve in panel **c**; for the black line it was set to the bottom curve. In all panels, $M = 100$ odors, $N = 400$ glomeruli, and three odors were presented on average.

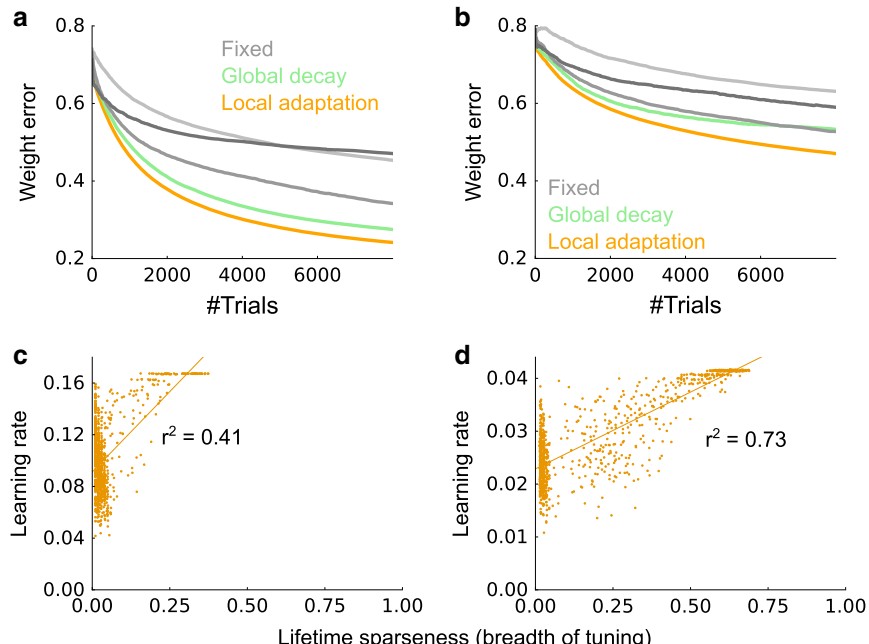

**Fig. 6 Adaptive synaptic plasticity. a** Weight error when $1/t\rho_j^t$ is fixed (gray lines), $\rho_j^t$ is fixed (light green), and fully adaptive (orange). For the gray lines we used learning rates of 0.01, 0.1, 1.0, correspond to light gray to dark gray. The sparsity, $c_o$, was 0.03. **b** Same as panel **a**, but with a lower sparsity, $c_o = 0.07$. **c, d** Correlations between the lifetime sparseness and the learning rate, after 300 stimuli were presented to the network, under more sparse (**c**: $c_o = 0.03$) and less sparse (**d**: $c_o = 0.07$) conditions. Lines are linear regressions, and each dot represents one granule cell. Correlation were significant for both **c** and **d** ($p \ll 10^{-6}$). Vertical clusters appearing on the left edges of the panels correspond to neurons with very small lifetime sparseness. In all panels, $M = 100$ odors, $N = 400$ glomeruli, and 3 (**a, c**) or 7 (**b, d**) odors were presented on average. Light-green and orange lines in **a** and **b** are mean over 50 simulations, while the rest were calculated from 10 simulations.

reflects an accumulation of information over time: at the beginning of learning, the olfactory stimuli contains a relatively large amount of information about the weights, and so the learning rate is large, and vice versa. The second term, $1/\rho_j^t$, is the cell-specific contribution to the learning rate. In steady state, it is given approximately by $1/\rho_j^t \propto 1/\langle c_j^2 \rangle_{odors}$ (the subscript "odors" indicates an average over odors).

It turns out that the second term is related to the lifetime sparseness, $S_j \equiv \langle c_j \rangle_{odors}^2 / \langle c_j^2 \rangle_{odors}$ (note that smaller $S_j$ means activity is more sparse; see subsection "Lifetime sparseness" in the Methods section and ref. [19]). Assuming the mean firing rate, $\langle c_j \rangle_{odors}$, is approximately constant (as we see in our simulations), then $1/\rho_j^t \propto S_j$. When the granule cells have broad, non-selective tuning, the lifetime sparseness is large, and the learning rate is

high; when the cells are sparse and have highly selective tuning, the lifetime sparseness is low, and so is the learning rate. Thus, if the mean granule cell responses are similar for all presented odors, the learning rate is large, encouraging neurons to modify their selectivity. If, on the other hand, the granule cell responses are sparse and selective, the learning rate is low, helping the neurons stabilize their acquired selectivity.

We examined the effects of the two factors—$1/t$ and $1/\rho_j^t$—on learning. When the learning rate, $1/t\rho_j^t$, was kept constant throughout learning, learning was slower, even when the learning rate was finely tuned (gray lines vs. orange line in Fig. 6a). This makes sense from a Bayesian perspective: early on, when weight uncertainty is large, learning should be fast (the dark gray line, which has the highest learning rate, drops rapidly), whereas after a large number of trials, when weight uncertainty is low, learning

should be slow (the lighter gray lines, which have lower learning rates, have better asymptotic performance). It is also consistent with the fine tuning required for the sparse coding model in Fig. 4a and b. When we fixed $1/\rho_j$ but included the global factor $1/t$, performance was better than the model with fixed learning rate (light-green vs. gray in Fig. 6a), yet still worse than the original fully adaptive model (light-green vs. orange in Fig. 6a). This was more clear under a less sparse setting ($c_o = 0.07$ in Fig. 6b, versus $c_o = 0.03$ in Fig. 6a). Furthermore, as predicted, we found that the learning rate of a cell, $1/t\rho_j^t$, is positively correlated with the lifetime sparseness at each time point (i.e. at fixed $t$) as shown in Fig. 6a and b. This correlation becomes weaker as the prior becomes more sparse (compare Fig. 6c and d, for which $c_o = 0.03$ and 0.07, respectively). That is because a very sparse prior (low $c_o$) helps the granule cells to be highly selective at an early stage, enabling the lifetime sparseness to quickly converge to a small value (vertical cluster on the left edge of Fig. 6c and d). These results indicate that the global and postsynaptic-neuron-specific adaptation of the learning rate cooperatively help fast learning.

**Learning concentration invariant representation and valence.** Our results so far indicate that olfactory learning is well characterized as an approximate Bayesian learning process. Our circuit estimates odor concentration, which is important for locating an odor source[32]. However, the perceived concentration depends on factors such as the distance from the odor source, its size, and wind speed. Thus, odor concentration is not a reliable indicator of the amount of reward expected. Hence, acquisition of a concentration-invariant representation is highly useful for many olfactory-guided behaviors.

A concentration-invariant representation is essentially a representation of the probability of an odor being present, denoted $\overline{p}_j$. Because of the spike in our prior, $\overline{p}_j = \Pr[c_j > 0]$, thus probability is easily decoded from M/T cells using the circuit depicted in Fig. 7a (see subsection "Learning of concentration-invariant representation" in the Methods section). Here, $\overline{p}_j$ could be represented in layer 2 of piriform cortex neurons, as that is the main downstream target of M/T cells, and odor representation in piriform cortex is approximately concentration-invariant[21,33]. As the granule cells acquire odor representation, neurons in piriform cortex acquire odor probability representation (cyan and dark blue line in Fig. 7e left).

While the circuit shown in Fig. 7a exhibits good performance, it is not consistent with the mammalian olfactory system, in two ways. First, the weights from the M/T cells to the granule cells have to be copied to the corresponding M/T to piriform cortex connections (i.e. $\mathbf{w}^P = \mathbf{w}^F$), something that is not biologically plausible. Second, a direct projection from granule cells to piriform cortex is needed, but such a connection does not exist. These inconsistencies can be circumvented by modifying the circuit heuristically (Fig. 7b–d). Weight copying can be avoided by learning $\mathbf{w}^P$ with local synaptic plasticity (Fig. 7b), although in the absence of the teaching signal from the granule cells, this naive extension does not work (dark blue line in Fig. 7e middle-left). However, introducing lateral inhibition among the piriform neurons (Fig. 7c) as observed experimentally[21], allows the piriform neurons to acquire odor representation (Fig. 7c and e middle-right), although the decoding performance was worse than the Bayesian model (Fig. 7e left vs. 7e middle-right). Finally, if connections from piriform cells to granule cells are added as well, the learning performance of granule cells became slightly better (Fig. 7d and e right), and more robust to changes in the strength of lateral inhibition (Fig. 7f). As expected, the responses of piriform neurons were mostly concentration-invariant (dark blue line in Fig. 7g), whereas granule cells showed a clear concentration dependence (cyan line in Fig. 7g). Thus, the architecture of the mammalian olfactory circuit indeed supports robust learning of concentration-invariant representation.

Once the circuit acquires a concentration-invariant representation, a circuit that performs odor–reward association can be constructed simply by taking the circuit depicted in Fig. 7d and adding a region that receives input from both piriform neurons and the reward system ($e_p$ in Fig. 7h). Olfactory tubercle could be the site for this odor–reward association[5,34], but it could be other regions, such as layer 3 of piriform cortex, as well. To test performance of this circuit, we implemented a go/no go task in which one odor is associated with a reward ($R = 1.0$), while another odor is associated with no reward ($R = 0.0$), regardless of concentrations. We simulated this task by randomly presenting rewarded or unrewarded stimulus with equal probability (see subsection "Go/no go task" in the Methods section). We used the circuit pre-trained with a large number of odors but without reward. When the reward prediction was learned with the projection from piriform cells, $\overline{p}$, to olfactory tubercle cells, $e_p$ (Fig. 7h), classification performance reaches 90% after just six trials (Fig. 7j; magenta lines). On the other hand, when the circuit learns the task directly from the glomeruli (Fig. 7i), though the circuit still learns to predict the reward as suggested previously[35], learning was much slower and the performance was worse even after a large amount of training (Fig. 7j; purple lines). After a dozen odor–reward association from piriform neurons, $\overline{p}$, olfactory tubercle cell activity, $e_p$, learned to represent the reward prediction given olfactory stimuli unless the concentration is very small (left half of Fig. 7k; in our model—$e_p$ is the reward prediction), and once the reward is presented at $\tau = 2.5\,\mathrm{s}$, the activity went back to near zero (right half of Fig. 7k; in our model, positive $e_p$ represents an error, and so drives learning).

These results indicate that unsupervised learning of odor representation may underlie fast reward-based learning, and the proposed Bayesian learning mechanism improves reward association by enabling robust odor representation in a data efficient way.

## Discussion

We formulated unsupervised olfactory learning in the mammalian olfactory system as a Bayesian optimization problem, then derived a set of local synaptic plasticity rules and neural dynamics that implemented Bayesian inference (Figs. 2 and 3). Our theory provides a normative explanation of the functional roles for the nonlinear transfer function and the developmental adaptation of the neuronal input gain (Fig. 5), both widely observed among sensory neurons. The model also predicts that the learning rate of dendro-dendritic connections should be approximately linear in the lifetime sparseness of the corresponding granule cells (Fig. 6). Finally, we extended the framework to learning of odor identity by piriform cortex, and showed that such learning supports rapid reward association (Fig. 7).

Our results suggest that adaptation of both input gain (Fig. 5) and learning rate (Fig. 6) are important for successful learning. The developmental reduction in input gain can be explained by a decrease in neural excitability, which is partially caused by the increased expression of $K^+$ channels[14]. Correspondingly, it is known that changes in channel expression at the dendrite modulate the sensitivity of synaptic plasticity[36]. In particular, it has been reported that elimination of voltage-gated $K^+$ channels enhances the induction of long-term potentiation[37]. These results suggest that developmental up-regulation of $K^+$ channel expression at the soma and the dendrite may underlie the adaptation of the input gain and learning rate.

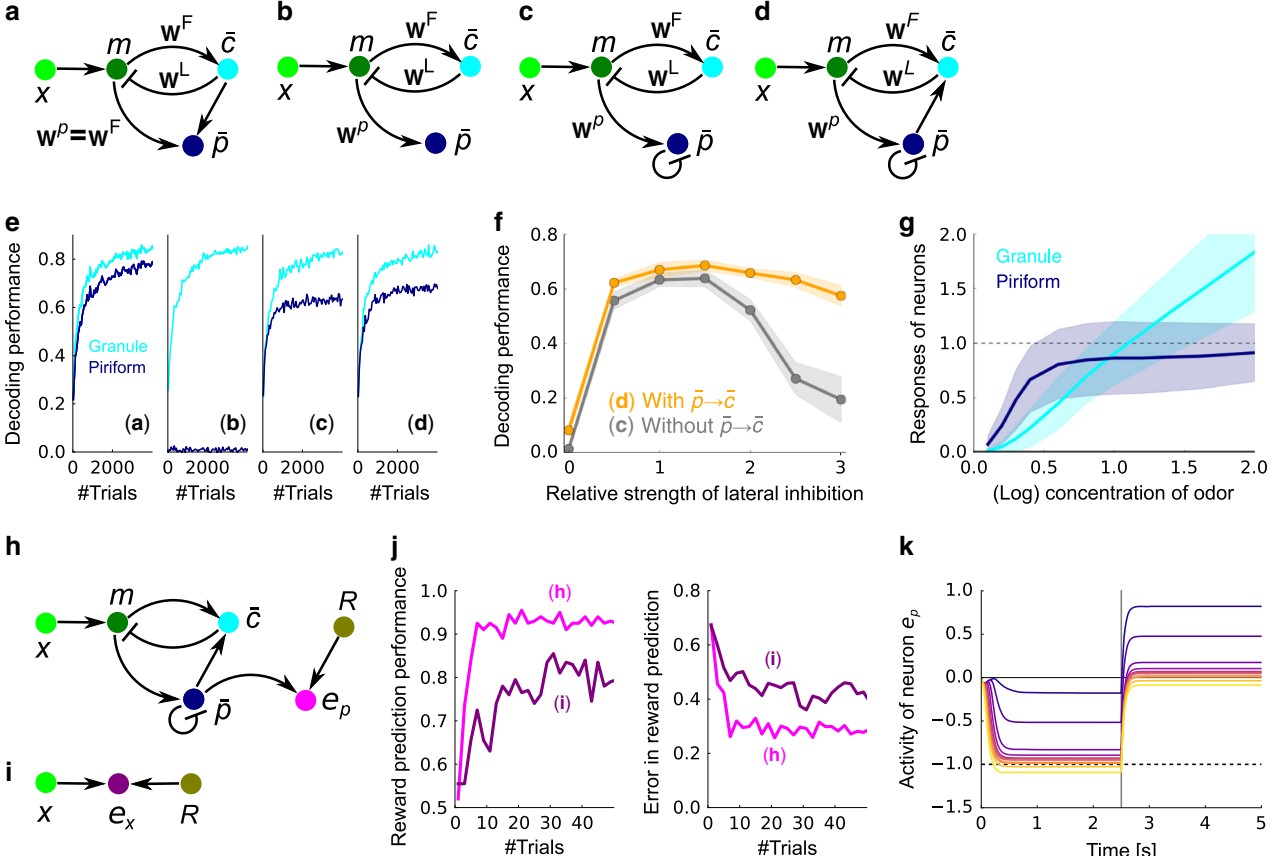

**Fig. 7 Learning a concentration invariant representation and an odor-reward association. a–d** A set of increasingly realistic decoding models. **a** The decoding model associated with our variational Bayesian inference algorithm. Note that the weights need to be copied from $\mathbf{w}^F$ to $\mathbf{w}^p$, something that is not biologically plausible. **b** Similar circuit, but with the mapping from $m$ to $\bar{p}$ learned via a local rule. **c** Same as **b**, but with lateral inhibition. **d** Same as **c**, but with feedback to the granule cells. **e** Learning performance for the models in **a–d** when decoding from granule cells (cyan) or piriform cortex (dark blue; see subsection "Odor estimation performance" in the "Methods" section). **f** Comparison of performance for model **c** (gray) and **d** (orange). Mean and standard deviation over 10 simulations are plotted. **g** Mean and standard deviation of responses of the granule cells, $\bar{c}$, and the piriform neurons, $\bar{p}$, for their selective odors presented at various concentrations. The responses were measured by presenting each odor in isolation with different concentration, and then averaging over populations. **h** Schematic of the reward prediction circuit utilizing concentration-invariant representation in the piriform cells, $\bar{p}$. **i** Direct reward prediction from neural activity at glomeruli. **j** Performance of odor–reward association measured by the classification performance (left) and the mean-squared error between the predicted reward and the actual reward (right) for the models in panels **h** (magenta) and **i** (purple). Lines are mean over 100 simulations. **k** The mean response of neuron $e_p$ given an odor associated with the reward. The vertical line at $\tau = 2.5$ s represents the reward presentation, and the dotted horizontal line is the sign-flipped reward value ($-R$). Different colors represents the different concentrations of the presented odor, from purple ($c \approx 0.1$) to yellow ($c \approx 2.0$). In all panels, $M = 50$ odors, $N = 200$ glomeruli, and three odors were presented on average, except for the go/no go task where one of two selected odors was presented randomly.

The cellular plasticity rules we derived explain multiple developmental changes in adult born granule cells. Experimentally, relative to young cells, mature granule cells have sparser selectivity[20], lower membrane resistance[17,18], and are less plastic[18], as predicted by our model. In addition, our results provide insight into the functional role of adult neurogenesis. As shown previously[8], if each synapse keeps track of its uncertainty, by removing the most uncertain synapses while adding synapses at a random position on the dendritic tree, a neuron can achieve sample-based Bayesian learning, making neurogenesis unnecessary. However, in our unsupervised learning framework, uncertainty is defined at neurons, not at synapses. As a result, from a Bayesian perspective, there is no good way to perform synaptogenesis. Thus, the brain should instead remove the most uncertain neurons, while at the same time randomly adding new ones.

The importance of the feedback circuit between M/T cells and granule cells has been noted previously[6,38], but plasticity mechanisms that generate this circuit have not been considered. Recently, several groups proposed learning algorithms for unsupervised olfactory learning using stochastic gradient descent[11,12,39], as in the case of our sparse coding model. However, as we have seen (Fig. 4), these algorithms are very unlikely to be fast. In addition to the sparse coding model, our problem setting is deeply related to Independent component analysis (ICA)[40]. Indeed, by using the sparseness as the measure of non-Gaussianity, unsupervised olfactory learning can be reformulated as an ICA problem[11].

The spike-and-slab prior employed here is widely used in machine learning[41], and has been applied to the sparse coding model of the early visual system[42], and a normative analysis of nonlinear transfer functions has been carried out previously[43]. A contribution of this work is the establishment of a link between the spike-and-slab prior and nonlinear transfer function of a neuron.

Studies of adaptive learning rates date back many decades[44,45]; more recent studies have taken a Bayesian approach to adaptive learning in simplified single neurons models[7]. In this study, we considered an unsupervised learning problem, and showed that the learning rate of excitatory feedforward connections should depend only on the postsynaptic activity, independent of the presynaptic activity. Moreover, our theory predicted a non-trivial relationship between the learning rate and the lifetime sparseness of the postsynaptic neuron (Fig. 6c and d).

Acceleration of reward-based learning by unsupervised learning (Fig. 7j) has been studied in the context of both semi-supervised learning and model-based reinforcement learning. In particular, the latter approach has been applied to rapid learning by animals, but these were limited to abstract models, not circuit-based implementations[46]. In the invertebrate literature, Bazhenov and colleagues (2013) studied the combination of unsupervised and reward-based learning in a computational model of the insect brain[47], but plasticity was applied only to the output connections (corresponding in our model to $\overline{p} \to e_p$ in Fig. 7h). Interestingly, in the invertebrate brain, the connections corresponding to $m \to \overline{p}$ are mostly random and fixed[48], so the acceleration shown in Fig. 7j is potentially unique to vertebrates.

While our approach gave us a model that is reasonably consistent with mammalian olfactory circuitry, it is not perfect. In particular, the architecture predicted by our approximate Bayesian algorithm does not match perfectly the architecture of the olfactory bulb, piriform cortex, and olfactory tubercle. We were able to make small modifications to our circuit so that it did match the biology, and still gave decent performance, but performance was about 10% worse than the circuit predicted purely by Bayesian inference (blue lines in Fig. 7e-left vs. 7e-right). This discrepancy between the predicted and observed architecture highlights a limitation of this approach, especially when applied to complex systems. In particular, it is difficult to include biological constraints, both because we do not know exactly what they are, and because there is no straightforward way to marry those constraints with a normative Bayesian approach. However, that is an important avenue for future work.

## Methods

### Stimulus configuration
On each trial, the response of the $i$th glomerulus is modeled as

$$x_i = \sum_j w_{ij} c_j + \sigma_x \xi_i \tag{6}$$

where $c_j$ is the concentration of odor $j$, and $\xi_i$ is a zero mean, unit variance Gaussian random variable. The Gaussian assumption is justified because, although olfactory sensory neurons fire with approximately Poisson statistics, 1000–10,000 sensory neurons converge to a single glomerulus[22], where OSN activity is conveyed to M/T cells as stochastic currents. We take the affinities, or mixing weights, $\mathbf{w}$, to be log normal, followed by a normalization step

$$\log \widetilde{w}_{ij} \sim \mathcal{N}(-\log(c_o M), 1) \tag{7a}$$

$$w_{ij} = \widetilde{w}_{ij} \times \frac{\frac{1}{NM} \sum_i^N \sum_j^M \widetilde{w}_{ij}}{\frac{1}{M} \sum_j^M \widetilde{w}_{ij}} \tag{7b}$$

where recall, $M$ is the number of odors and $N$ is the number of glomeruli. The factor multiplying $\widetilde{w}_{ij}$ is 1 on average, so the normalization step does not have a huge effect on the weights. However, it forces $\sum_j w_{ij}$ to be strictly independent of $i$, which makes the learning process less noisy.

On each trial, odors $c_j$ ($j = 1, 2, \ldots, M$) are generated from the spike-and-slab prior given as

$$p_c(c_j) = (1 - c_o)\delta(c_j) + c_o \frac{\alpha^\alpha}{\Gamma(\alpha)} c_j^{\alpha-1} e^{-\alpha c_j} \Theta(c_j), \tag{8}$$

where $\Theta(x)$ is a Heaviside function. We used $\alpha = 3$ everywhere except Supplementary Fig. 1, where we used $\alpha = 1$. Under this prior, each odor is independently presented with probability $c_o$, and its amplitude follows a Gamma distribution with unit mean (Fig. 1a left). Note that the amplitude, $c_j$, reflects

log-concentration rather than concentration[24]. To avoid the null stimulus, we resampled the odors if all of the $c_j$ were 0 on any particular trial.

### Bayesian model
As discussed in the main text, we mainly focus on unsupervised learning, in which animals see only glomeruli activity and must make sense of it. This is essentially a clustering problem: if the same pattern of glomeruli activity occurs multiple times, the brain should recognize it as an odor. The activity patterns at the glomeruli are determined by the product of odorant concentrations in the inhaled air, and the affinities of the OSNs for those odorants. Thus, to recognize an odor, animals have to effectively learn the affinities of OSNs for each odor, and store them in the olfactory circuitry. As we will see, in our model they are stored as weights between M/T cells and granule cells. Once those weights are stored, if an odor co-occurs with a reward (or punishment), the valance of that odor can be determined. And indeed, we find that unsupervised learning enables rapid learning of odor–reward associations.

More formally, the goal of the olfactory system is to infer the odor at time $t$, $\mathbf{c}_t$, given all past presentations of odors, $\mathbf{x}_{1:t} \equiv \{\mathbf{x}_1, \mathbf{x}_2, \ldots, \mathbf{x}_t\}$. Because the weights are not known, they must be integrated out

$$p(\mathbf{c}_t | \mathbf{x}_{1:t}) = \int d\mathbf{w}\, p(\mathbf{c}_t, \mathbf{w} | \mathbf{x}_{1:t}). \tag{9}$$

Using Bayes' theorem, this can be written in a more intuitive form

$$p(\mathbf{c}_t | \mathbf{x}_{1:t}) \propto \int d\mathbf{w}\, p(\mathbf{x}_t | \mathbf{c}_t, \mathbf{w}) p_c(\mathbf{c}_t) p(\mathbf{w} | \mathbf{x}_{1:t-1}) \tag{10}$$

where, recall, $p_c(\mathbf{c}_t)$ is the prior over odors. To derive this expression, we used two facts: given $\mathbf{c}_t$ and $\mathbf{w}$, $\mathbf{x}_t$ does not depend on past observations, and $\mathbf{c}_t$ does not depend on past observations. The first term on the right-hand side, $p(\mathbf{x}_t | \mathbf{c}_t, \mathbf{w})$ is the likelihood given the weights; but because we do not know the weights, we have to marginalize over them given past observations. The marginalization step is intractable, as we have to introduce past odors and then integrate them out. This leaves us with an integral over $\mathbf{w}$ (Eq. (10)) that cannot be performed analytically. And even if it could, the circuit would have to memorize all past stimuli, $\mathbf{x}_1, \mathbf{x}_2, \ldots, \mathbf{x}_{t-1}$. We thus have to perform approximate inference. For that we make a variational approximation.

### Variational approximation
The integral in Eq. (10) becomes easier if the distributions factorize. We thus make the variational approximation

$$p(\mathbf{c}, \mathbf{w} | \mathbf{x}_{1:t-1}, \mathbf{x}) \approx q^t(\mathbf{w}, \mathbf{c}) \equiv \prod_{ij} q_{ij}^{w,t}(w_{ij}) \times \prod_j q_j^c(c_j) \tag{11}$$

where, to avoid a proliferation of subscripts, we suppress the fact that $\mathbf{c}$ and $q_j^c$ are to be evaluated at trial $t$; in line with this, to simplify subsequent equations we replace $\mathbf{x}_t$ with $\mathbf{x}$; and, as is standard, we suppress the dependence of $q$ on $\mathbf{x}_{1:t}$.

The variational distributions, $q_{ij}^{w,t}$ and $q_j^c$, are found by minimizing the KL-divergence with respect to the true distribution, with the KL-divergence given by

$$D_{KL}[q^t(\mathbf{w}, \mathbf{c}) || p(\mathbf{c}, \mathbf{w} | \mathbf{x}_{1:t-1}, \mathbf{x})] = \int d\mathbf{c} d\mathbf{w}\, q^t(\mathbf{w}, \mathbf{c}) \log \frac{q^t(\mathbf{w}, \mathbf{c})}{p(\mathbf{c}, \mathbf{w} | \mathbf{x}_{1:t-1}, \mathbf{x})}. \tag{12}$$

As is straightforward to show[9], minimizing this quantity leads to the update rules

$$\log q_{ij}^{w,t}(w_{ij}) \sim \langle \log p(\mathbf{x} | \mathbf{c}, \mathbf{w}) \rangle_{\backslash w_{ij}} + \langle \log p(\mathbf{w} | \mathbf{x}_{1:t-1}) \rangle_{\backslash w_{ij}} \tag{13a}$$

$$\log q_j^c(c_j) \sim \langle \log p(\mathbf{x} | \mathbf{c}, \mathbf{w}) \rangle_{\backslash c_j} + \log p_c(c_j) \tag{13b}$$

where $\sim$ indicates equality up to a constant, $\backslash w_{ij}$ indicates an average with respect to the variational distribution over all variables except $w_{ij}$, and, similarly, $\backslash c_j$ indicates an average with respect to the variational distribution over all variables except $c_j$. In the first equation, we approximate $p(\mathbf{w} | \mathbf{x}_{1:t-1})$ with the variational distribution at the previous time step, $\prod_{ij} q_{ij}^{w,t-1}(w_{ij})$, which makes the marginalization self-consistent. This approximation breaks down early in the learning process; nevertheless, in practice it works quite well. Using this approximation, we arrive at

$$q_{ij}^{w,t}(w_{ij}) \propto q_{ij}^{w,t-1}(w_{ij}) \exp\left[\langle \log p(\mathbf{x} | \mathbf{c}, \mathbf{w}) \rangle_{\backslash w_{ij}}\right] \tag{14a}$$

$$q_j^c(c_j) \propto p_c(c_j) \exp\left[\langle \log p(\mathbf{x} | \mathbf{c}, \mathbf{w}) \rangle_{\backslash c_j}\right]. \tag{14b}$$

In the next two subsections we derive explicit update rules by computing the averages in these expressions.

### The variational odor distribution
To find the variational distribution over odors, we need to compute the average over $\log p(\mathbf{x} | \mathbf{c}, \mathbf{w})$ that appears on the right-hand

side of Eq. (14b). Using the fact that the $\mathbf{x}$ follows a Gaussian distribution, we have

$$
\langle \log p(\mathbf{x}_t | \mathbf{c}, \mathbf{w}) \rangle_{\backslash c_j} \sim -\frac{1}{2\sigma_x^2} \left\langle \sum_i \left( x_i^t - \sum_m w_{im} c_m \right)^2 \right\rangle_{\backslash c_j}
$$

$$
\sim -\frac{\sum_i \langle w_{ij}^{t\,2} \rangle}{2\sigma_x^2} \left( c_j - \frac{1}{\sum_i \langle w_{ij}^{t\,2}\rangle} \sum_i \langle w_{ij}^t \rangle \left[ x_i^t - \sum_{m \neq j} \langle w_{im}^t \rangle \langle c_m \rangle \right] \right)^2,
$$

$$(15)$$

where the averages are with respect to the variational distribution. This is Gaussian, and it is straightforward to work out the mean and variance. Note that both depend on the first and second moments of the weights (which, as we will see below, determine the variational weight distribution) evaluated, importantly, at time $t$. However, synaptic plasticity is much slower than neural dynamics, so it is reasonable to update the weights on a slower timescale than concentration. Thus, when evaluating the mean and variance, we use the weight distribution on the previous time step. Using $\mu_j^t$ and $1/\lambda_j^t$ to denote the mean and variance, and making this approximation, we have

$$
\mu_j^t \equiv \frac{1}{\sum_i \langle w_{ij}^{t-1\,2}\rangle} \sum_i \langle w_{ij}^{t-1}\rangle \left[ m_i^t + \langle w_{ij}^{t-1}\rangle \langle c_j\rangle \right] \tag{16a}
$$

$$
\lambda_j^t \equiv \frac{1}{\sigma_x^2} \sum_i \langle w_{ij}^{t-1\,2}\rangle \tag{16b}
$$

where we made the definition

$$
m_i^t \equiv x_i^t - \sum_{j=1}^M \langle w_{ij}^{t-1}\rangle \langle c_j\rangle . \tag{17}
$$

The distribution $q_j^c(c_j)$ can now be written in a very compact form

$$
q_j^c(c_j) \propto p_c(c_j) \; \exp\left[ -\frac{\lambda_j^t}{2} \left( c_j - \mu_j^t \right)^2 \right]. \tag{18}
$$

As we will see below, to update the weights we just need the first and second moments of $c_j$ (see Eq. (27a)). And for the reward-based learning, we need the probability that $c_j$ is positive. These quantities are straightforward, if tedious, to compute, and are given as follows.

For the first moment,

$$
\langle c_j\rangle = \frac{1}{Z_j\sqrt{\lambda_j}} \left( [2 + \alpha_j^2] + \alpha_j[3 + \alpha_j^2]\Psi(\alpha_j) \right), \tag{19}
$$

where the average is with respect to the distribution in Eq. (18), $Z_j$ is the normalization constant

$$
Z_j \equiv \frac{2(1 - c_o)}{27 c_o} \lambda_j^{3/2} + \alpha_j + (1 + \alpha_j^2)\Psi(\alpha_j), \tag{20}
$$

and $\alpha_j$ and $\Psi(\alpha_j)$ are defined by

$$
\alpha_j \equiv \sqrt{\lambda_j}\mu_j - \frac{3}{\sqrt{\lambda_j}} \tag{21a}
$$

$$
\Psi(\alpha_j) \equiv \sqrt{2\pi} e^{\alpha_j^2/2} \Phi(\alpha_j), \tag{21b}
$$

with $\Phi$ the cumulative normal function

$$
\Phi(\alpha) \equiv \frac{1}{\sqrt{2\pi}} \int_{-\infty}^{\alpha} e^{-x^2/2} dx. \tag{22}
$$

Similarly, the second moment is given by

$$
\langle c_j^2\rangle = \frac{1}{Z_j\lambda_j} \left( \alpha_j(5 + \alpha_j^2) + (3 + 6\alpha_j^2 + \alpha_j^4)\Psi(\alpha_j) \right). \tag{23}
$$

And finally, the probability that an odor is present is written

$$
\Pr[c_j > 0] = \frac{1}{Z_j} \left( \alpha_j + (1 + \alpha_j^2)\Psi(\alpha_j) \right) . \tag{24}
$$

**The variational weight distribution.** To find the variational distribution over weights, we need to compute the average on the right-hand side of Eq. (14a). This is the same as Eq. (15), except that the average now excludes $w_{ij}$ rather than $c_j$,

$$
\langle \log p(\mathbf{x}|\mathbf{c}, \mathbf{w}) \rangle_{\backslash w_{ij}} \sim -\frac{1}{2\sigma_x^2} \left\langle \left( x_i - \sum_m w_{im} c_m \right)^2 \right\rangle_{\backslash w_{ij}}
$$

$$
\sim -\frac{\langle c_j^2\rangle}{2\sigma_x^2} \left( w_{ij} - \frac{\langle c_j\rangle}{\langle c_j^2\rangle} \left[ x_i - \sum_{m \neq j} \langle w_{im}^t\rangle \langle c_m\rangle \right] \right)^2 \tag{25}
$$

where the averages are, as above, with respect to the variational distributions. This is a quadratic function of $w_{ij}$; thus, if we assume that $q_{ij}^{w,t-1}(w_{ij})$ is Gaussian, then

$q_{ij}^{w,t}(w_{ij})$ is also Gaussian. Using $\overline{w}_{ij}^t$ and $1/(t\rho_j^t)$ to denote the mean and variance at time $t$, respectively (the latter to anticipate the $1/t$ falloff of the variance expected under Bayesian filtering), Eq. (14a) becomes

$$
-\frac{t\rho_j^t}{2}\left(w_{ij} - \overline{w}_{ij}^t\right)^2 \sim -\frac{(t-1)\rho_j^{t-1}}{2}\left(w_{ij} - \overline{w}_{ij}^{t-1}\right)^2 - \frac{\langle c_j^2\rangle}{2\sigma_x^2}\left(w_{ij} - \frac{\langle c_j\rangle}{\langle c_j^2\rangle}\left[x_i - \sum_{m\neq j}\overline{w}_{im}^t\langle c_m\rangle\right]\right)^2. \tag{26}
$$

As in Eq. (15), $\overline{w}^t$ appears on the right-hand side of Eq. (26). However, very fast synaptic plasticity is required for solving this equation recursively for all the weights. We thus approximate the right-hand side by using the previous timestep, $t-1$, rather than the current one, $t$; an approximation that should be good when the weights change slowly. Doing that, we arrive at the update rules

$$
\rho_j^t = (1 - 1/t)\rho_j^{t-1} + \frac{1/t}{\sigma_x^2}\langle c_j^2\rangle \tag{27a}
$$

$$
\overline{w}_{ij}^t = (1 - 1/t)\frac{\rho_j^{t-1}}{\rho_j^t}\overline{w}_{ij}^{t-1} + \frac{1/t}{\rho_j^t\sigma_x^2}\langle c_j\rangle\left(m_i^t + \overline{w}_{ij}^{t-1}\langle c_j\rangle\right) \tag{27b}
$$

where we used Eq. (17) to simplify the second expression. Note that the update rule for $\overline{w}_{ij}^t$ is local, as it depends only on variables indexed by $i$ and $j$. The update rule for $\rho_j^t$ is also local, and in fact depends only on variables indexed by $j$.

Finally, it is convenient to write the update rules for the mean and precision of the variational distribution over concentration, Eq. (16), in terms of $\overline{w}_{ij}$ and $\rho_j$,

$$
\mu_j^t \equiv \frac{1}{\sigma_x^2\lambda_j^t}\sum_i \overline{w}_{ij}^{t-1}\left[m_i^t + \overline{w}_{ij}^{t-1}\langle c_j^t\rangle\right] \tag{28a}
$$

$$
\lambda_j^t \equiv \frac{1}{\sigma_x^2}\sum_i \left(\overline{w}_{ij}^{t-1}\right)^2 + \frac{N}{\sigma_x^2(t-1)\rho_j^{t-1}} . \tag{28b}
$$

As shown in Fig. 5c, the transfer function shifts to the right with learning. This seems counter-intuitive: because the weights become more certain with learning, it should take less input to the granule cells to produce activity; this suggests that the transfer functions should shift left, not right. However, an increase in certainty is not the only thing that changes with learning; the weights also become more diverse, capturing the diverse responses of glomeruli for each odor. The diversity increases the variance of the input to the granule cells, and so to ensure a sparse response with increasing diversity, the transfer functions need to shift to the right. In our model, increased diversity (the first term in Eq. (28b)) had a larger effect than increased certainty (the second term), resulting in a net rightward shift in the transfer functions.

**Network model.** The analysis in the previous sections revealed that under the variational approximation, the distribution of the odors and the weights are updated locally. Thus, we implement the update rules in a network model of the olfactory bulb. The update of the weight distribution, $q_{ij}^{w,t}(w_{ij})$, depends on $\langle c_j\rangle$ and $\langle c_j^2\rangle$, as shown in Eq. (27), while the update of the odor distribution, $q_j^{c,t}(c_j)$, depends on $\overline{w}_{ij}$ and $\rho_j$, as shown in Eq. (28). Ideally, all these parameters should be updated simultaneously. However, as mentioned above, updates to synaptic weights are typically much slower than the neural dynamics, so here we consider a two step update. First, the relevant parameters of the variational odor distribution, $\langle c_j\rangle$ and $\langle c_j^2\rangle$, are updated using the mean and precision of the weight distribution, $\overline{w}_{ij}$ and $\rho_j$, evaluated at $t-1$. Then, $\overline{w}_{ij}$ and $\rho_j$ are updated using the first and second moments of the weights, $\langle c_j\rangle$ and $\langle c_j^2\rangle$, evaluated at time $t$.

**Neural dynamics.** Our goal is to write down a set of dynamical equations for $\langle c_j\rangle$ and $\langle c_j^2\rangle$ whose fixed points correspond to the values given in Eqs. (19) and (23), respectively. Examining these equations, we see that $\langle c_j\rangle$ and $\langle c_j^2\rangle$ depend on $\alpha_j$ and $\lambda_j$; after a small amount of algebra (involving the insertion of Eq. (28a) into Eq. (21a)), $\alpha_j$ may be written

$$
\alpha_j = \frac{1}{\sqrt{\lambda_j}\sigma_x^2}\left(\sum_i \overline{w}_{ij}m_i + \sum_i \overline{w}_{ij}^2\langle c_j\rangle - 3\sigma_x^2\right) . \tag{29}
$$

To avoid clutter, we dropped the dependence on time, but the weights should be evaluated at time $t-1$ and all other variables at time $t$.

Because neither $\alpha_j$ nor $\lambda_j$ (the latter given in Eq. (28b)) depend on $\langle c_j^2\rangle$, we can write down coupled equations for $\langle c_j\rangle$ and $m_i$; the solution of those equations gives us the values of $\alpha_j$ and $\lambda_j$, which in turn gives us, via Eq. (23), $\langle c_j^2\rangle$. Using, for notational ease, $\overline{c}_j$ rather than $\langle c_j\rangle$, the simplest such equations (derived from Eqs. (17) and (19)) are

$$
\tau_r \frac{dm_i}{d\tau} = x_i - m_i - \sum_{j=1}^M w_{ij}^L \overline{c}_j \tag{30}
$$

$$
\tau_r \frac{d\overline{c}_j}{d\tau} = -\overline{c}_j + F_j\left[\sum_{i=1}^N w_{ji}^F m_i; \overline{c}_j\right] \tag{31}
$$

where $\tau_r$ is the time constant of the firing rate dynamics, and the nonlinear transfer function, $F$, is given by the right-hand side of Eq. (19)

$$F_j\left[\sum_{i=1}^N w_{ji}^{\mathrm{F}} m_i; \overline{c}_j\right] \equiv \frac{1}{\sqrt{\lambda_j}} \frac{(2+\alpha_j^2)+\alpha_j(3+\alpha_j^2)\Psi(\alpha_j)}{\frac{2(1-c_o)}{27c_o}\lambda_j^{3/2}+\alpha_j+(1+\alpha_j^2)\Psi(\alpha_j)} \quad (32)$$

with $\alpha_j$ given in Eq. (29) and $\lambda_j$ in Eq. (28b). Note that we have replaced the average weights, $\overline{w}_{ij}$, with two different weights, $w_{ij}^{\mathrm{L}}$ and $w_{ij}^{\mathrm{F}}$. Ideally, we should have $w_{ji}^{\mathrm{F}} = w_{ij}^{\mathrm{L}} = \overline{w}_{ij}$, but, for biological plausibility, we allow these reciprocal synapses to be learned independently. Note that when evaluating $\alpha_j$, Eq. (29), $w_{ij}^{\mathrm{F}}$ should be used. Although the expression for $F_j$ seems complicated, the transfer functions are relatively smooth, and resemble experimentally observed ones (see Fig. 5).

As shown in Fig. 3b, this dynamical system resembles the neural dynamics of the olfactory bulb, under the assumption that $m_i$ and $\overline{c}_j$ are the firing rates of M/T cells and the granule cells, respectively. With this assumption, $w_{ji}^{\mathrm{F}}$ is the connection from M/T cell $i$ to granule cell $j$ and $w_{ij}^{\mathrm{L}}$ is the connection from granule cell $j$ to M/T cell $i$.

Finally, the second moment of the concentration is given, via Eq. (23), by

$$\langle c_j^2\rangle = G_j\left[\sum_i^N w_{ji}^{\mathrm{F},t-1} m_i; \overline{c}_j\right] \equiv \frac{1}{\lambda_j} \frac{\alpha_j(5+\alpha_j^2)+(3+6\alpha_j^2+\alpha_j^4)\Psi(\alpha_j)}{\frac{2(1-c_o)}{27c_o}\lambda_j^{3/2}+\alpha_j+(1+\alpha_j^2)\Psi(\alpha_j)} . \quad (33)$$

**Synaptic plasticity**. After trial $t$, the average feedforward weights, $w_{ji}^{\mathrm{F}}$, and the average lateral weights, $w_{ij}^{\mathrm{L}}$, are updated as in Eq. (27b)

$$w_{ji}^{\mathrm{F},t} = \left(1-\delta_j^{w,t}\right)w_{ji}^{\mathrm{F},t-1} + \frac{1/t}{\rho_j^t \sigma_x^2}\overline{c}_j m_i \quad (34a)$$

$$w_{ij}^{\mathrm{L},t} = \left(1-\delta_j^{w,t}\right)w_{ij}^{\mathrm{L},t-1} + \frac{1/t}{\rho_j^t \sigma_x^2} m_i \overline{c}_j \quad (34b)$$

$$\delta_j^{w,t} \equiv \frac{1}{t} + \left(1-\frac{1}{t}\right)\left(1-\frac{\rho_j^{t-1}}{\rho_j^t}\right) - \frac{\overline{c}_j^2}{t\rho_j^t \sigma_x^2} . \quad (34c)$$

We used the firing rates $m_i$ and $\overline{c}_j$ at the end of trial, after the neural dynamics has reached steady state. As the weight updates depend primarily on the product of $m_i$ and $\overline{c}_j$, the learning rules are essentially Hebbian. Note that if the initial conditions are the same (i.e., if $w_{ji}^{\mathrm{F},0} = w_{ij}^{\mathrm{L},0}$), then $w_{ji}^{\mathrm{F},t}$ and $w_{ij}^{\mathrm{L},t}$ will remain the same for all time. This is reasonable given that connections between M/T cells and granule cells are dendro-dendritic.

The variance of the weights, $1/t\rho_j^t$, consists of two components. The first, $1/t$, represents the global hyperbolic decay in the learning rate due to accumulation of information. In our simulations, we started $t$ from $t = t_{\min}$ to suppress the influence of the initial samples; this is equivalent to using a trial-dependent discount factor $1/(t+t_{\min})$ instead of $1/t$, where $t$ is the actual trial count. The second, $\rho_j^t$, represents the neuron-specific contribution to the precision, and is given, via Eqs. (27) and (23), by

$$\rho_j^t = (1-1/t)\rho_j^{t-1} + \frac{1}{t\sigma_x^2}G_j\left[\sum_i^N w_{ji}^{\mathrm{F},t-1} m_i; \overline{c}_j\right] , \quad (35)$$

where $G_j$, the second moment of the concentration, is given in Eq. (33).

**Models with various priors on odor concentration**. In our model setting, the prior over concentration, $p_c(c)$, enters via Eq. (14b), and affects the transfer functions $F$ and $G$, given in Eqs. (32) and (33), respectively. Choosing different priors gives different transfer function. Below we consider two common ones: non-negative, and non-negative with an exponential decay.

The first of these is actually an improper prior, $p_c(c) \propto \Theta(c)$. This results in gain functions of the form

$$F[\mu_j; \lambda_j] = \mu_j + \frac{1}{\sqrt{\lambda_j}\Psi\left[\sqrt{\lambda_j}\mu_j\right]} \quad (36a)$$

$$G[\mu_j; \lambda_j] = \mu_j F[\mu_j; \lambda_j] + \frac{1}{\lambda_j} \quad (36b)$$

where $\mu_j$ and $\lambda_j$ are given in Eqs. (28a) and (28b), respectively.

Under the non-negative prior introduced above, all positive concentrations are equally likely. However, that is not the case in a typical environment. Far more realistic is to assume that large concentrations are exponentially unlikely, yielding a prior of the form $p_c(c) = \frac{1}{c_o}\exp(-c/c_o)$. (The decay constant, $c_o$, was chosen so that the mean is equal to $c_o$, the same mean as in the true generative model.) For this

prior, the functions $F$ and $G$ are

$$F[\mu_j; \lambda_j] = \left(\mu_j - \frac{1}{c_o\lambda_j}\right) + \frac{1}{\sqrt{\lambda_j}\Psi\left[\sqrt{\lambda_j}\mu_j - \frac{1}{c_o\sqrt{\lambda_j}}\right]} \quad (37a)$$

$$G[\mu_j; \lambda_j] = \left(\mu_j - \frac{1}{c_o\lambda_j}\right)F[\mu_j; \lambda_j] + \frac{1}{\lambda_j} . \quad (37b)$$

While this prior is suboptimal for olfactory learning, experimental results from visual cortex indicate that the transfer function there resembles the one in Eq. (37a)[49] (black curve in Fig. 5a). Indeed, in early visual regions, where the prior is arguably more continuous[10], this shifted rectified-linear transfer function, might be more beneficial[50].

**Learning concentration invariant representations**. Up to now we focused on the expected concentration, $\overline{c}_j$. However, in natural environments animals often care more about whether or not an odor exists in its vicinity than what its concentration is. From a Bayesian perspective, this means the animals should compute the probability that an odor is present, denoted $\overline{p}_j$. Using Eq. (24), $\overline{p}_j$ can be estimated as the steady state of the following dynamics:

$$\tau_r \frac{d\overline{p}_j}{d\tau} = -\overline{p}_j + H_j\left[\sum_i w_{ji}^{\mathrm{F}} m_i, \overline{c}_j\right] \quad (38)$$

where $H_j$, which is approximately sigmoidal, is given, via Eq. (24), by

$$H_j\left[\sum_i w_{ij}^{\mathrm{F}} m_i, \overline{c}_j\right] = \frac{\alpha_j + (1+\alpha_j^2)\Psi(\alpha_j)}{\frac{2(1-c_o)}{27c_o}\lambda_j^{3/2}+\alpha_j+(1+\alpha_j^2)\Psi(\alpha_j)} \quad (39)$$

with $\alpha_j$ given in Eq. (29), but with $\overline{w}_{ij}$ replaced by $w_{ij}^{\mathrm{F}}$ in that equation as before.

In principle, neurons receiving input, $m_i$, from M/T cells, such as layer 2 piriform cortex neurons, can decode the odor probability, as shown in Fig. 7a and 7e-left. However, to calculate $H_j$ given input from M/T cells, the neuron would need to know the weights, $w_{ij}^{\mathrm{F}}$, as well as $\lambda_j$ and $\overline{c}_j$ (the latter because $\alpha_j$ depends on $\overline{c}_j$; see Eq. (29)). This is clearly unrealistic, because there is no known biological mechanism that enables copying weights. Moreover, because granule cells do not have output projections, except for the dendro-dendritic connections with M/T cells, piriform neurons cannot know $\overline{c}_j$ directly. Nevertheless, piriform neurons can learn to decode the concentration-invariant representation, $\overline{p}_j$, as follows.

Let us use $w_{ji}^{\mathrm{p}}$ to denote the mean weight from M/T cells to the piriform neurons (see Fig. 7b–d). Assume for the moment that $w_{ji}^{\mathrm{p}} \approx w_{ji}^{\mathrm{F}}$; shortly we will write down a learning rule that achieves this (see Eq. (43)). This takes care of the weights, but we also need an approximation to $\overline{c}_j$. For that, we notice that if the estimation is unbiased, on average both $\overline{c}_j$ and $\overline{p}_j$ are equal to $c_o$. Thus, the simplest way to approximate $\overline{c}_j$ with the information available to the $j$th piriform neuron is to use $\overline{c}_j \approx \overline{p}_j$. Under this approximation, and using $w_{ji}^{\mathrm{p}}$ in place of $w_{ji}^{\mathrm{F}}$, Eq. (38) becomes

$$\tau_r \frac{d\overline{p}_j}{d\tau} = -\overline{p}_j + H_j\left[\sum_i w_{ij}^{\mathrm{p}} m_i, \overline{p}_j\right] \quad (40)$$

where $H_j$ is the same as Eq. (39), but with $\alpha_j$ replaced by $\alpha_j^{\mathrm{p}}$—the analog of $\alpha_j$, but with $w_{ij}^{\mathrm{p}}$ and lateral inhibition

$$\alpha_j^{\mathrm{p}} \equiv \frac{1}{\sqrt{\lambda_j^{\mathrm{p}}\sigma_x^2}}\left(\sum_i w_{ij}^{\mathrm{p}} m_i + \sum_i \left(w_{ij}^{\mathrm{p}}\right)^2 \overline{p}_j - \sigma_x^2\left[3 + \lambda_j^{\mathrm{p}}\sum_{k\neq j} J_{jk}\overline{p}_k\right]\right) \quad (41)$$

where considering the analogy with Eq. (28b), $\lambda_j^{\mathrm{p}}$ is given by

$$\lambda_j^{\mathrm{p}} \equiv \frac{1}{\sigma_x^2}\sum_{i=1}^N \left(w_{ji}^{\mathrm{p}}\right)^2 + \frac{N}{\sigma_x^2(t-1)\rho_j^{\mathrm{p},t-1}} . \quad (42)$$

As above, $\overline{p}_j$ evolves with the weights set to their values updated at the end of previous trial. Once the neural dynamics reaches steady state, the weights are updated as in Eq. (34)

$$w_{ji}^{\mathrm{p},t} = \left(1-\delta_j^{\mathrm{p},t}\right)w_{ji}^{\mathrm{p},t-1} + \frac{1/t}{\rho_j^{\mathrm{p},t}\sigma_x^2}F_j\left[\sum_i^N w_{ji}^{\mathrm{p},t-1} m_i, \overline{p}_j\right] m_i \quad (43a)$$

$$\delta_j^{\mathrm{p},t} \equiv \frac{1}{t} + \left(1-\frac{1}{t}\right)\left(1-\frac{\rho_j^{\mathrm{p},t-1}}{\rho_j^{\mathrm{p},t}}\right) - \frac{1}{t\rho_j^{\mathrm{p},t}\sigma_x^2}\left(F_j\left[\sum_i^N w_{ji}^{\mathrm{p},t-1} m_i, \overline{p}_j\right]\right)^2 \quad (43b)$$

and the precision as in Eq. (27a)

$$\rho_j^{\mathrm{p},t} = (1-1/t)\rho_j^{\mathrm{p},t-1} + \frac{1/t}{\sigma_x^2}G_j\left[\sum_i^N w_{ji}^{\mathrm{p},t-1} m_i, \overline{p}_j\right]. \quad (44)$$

Here $F_j$ and $G_j$ are the estimated first/second moment given in Eqs. (32) and Eq.

(33), but calculated with $\alpha_j^p$ in Eq. (41). In steady state, these two terms approximate $\bar{c}_j$ and $\langle c_j^2 \rangle$, respectively. In addition, to ensure sparse piriform cell firing[51], we introduced Hebbian plasticity to the lateral weights $J_{jk}$,

$$\Delta J_{jk} = 0.1 \bar{p}_k \left( -5 c_o J_{jk} + \bar{p}_j \right) , \tag{45}$$

while bounding $J_{jk} > 0$ and enforcing $J_{jj} = 0$. We initialized $J_{jk}$ by $J_{jk} = 0.02$.

In Fig. 7e (panel d), 7f (orange line), 7g, and 7j–k, we modified the transfer function $F_j$ of granule cells by replacing the prior term $c_o$ with the input from piriform neuron $\bar{p}_j$. This means that $F_j^D$ is written as

$$F_j^D \left[ \sum_{i=1}^{N} w_{ji}^F m_i; \bar{c}_j, \bar{p}_j \right] \equiv \frac{1}{\sqrt{\lambda_j}} \frac{(2 + \alpha_j^2) + \alpha_j(3 + \alpha_j^2)\Psi(\alpha_j)}{\frac{2(1-\bar{p}_j)}{27 \bar{p}_j} \left( \lambda_j \right)^{3/2} + \alpha_j + (1 + \alpha_j^2)\Psi(\alpha_j)} \tag{46}$$

where $\alpha_j$ is still given by Eq. (29). We modulated the gain function $G_j$ of granule cells, Eq. (33), in the same way, by replacing $c_o$ with $\bar{p}_j$. In Fig. 7f, we changed the relative strength of lateral inhibition by replacing $J_{jk}$ in Eq. (41) with $\kappa_J J_{jk}$ where $\kappa_J$, the relative strength, ranged from 0 to 3, as shown in the $x$-axis of Fig. 7f, while using the original $J_{jk}$ for the weight update.

**Reward-based learning.** Assuming that the reward amplitude depends only on the identity of the odors, not on their concentrations, the reward, $R$, on trial $t$ is given by

$$R = \sum_{j=1}^{M} a_j \Theta(c_j) + \sigma_\zeta \zeta_t \tag{47}$$

where $\zeta_t$ is a zero mean, unit variance Gaussian random variable, and $\Theta(x)$ is a Heaviside function.

To estimate the reward, we augment the circuit in Fig. 7d by introducing a set of neurons, denoted $e_p$, that receive input both from $\bar{p}_j$ and the reward, $R$ (see Fig. 7h). Using $\bar{a}$ to denote those weights, the natural neural dynamics of $e_p$ is

$$\tau_r \frac{de_p}{d\tau} = -e_p + \hat{R}_t(\tau) - \sum_j \bar{a}_j \bar{p}_j. \tag{48}$$

To represent the delay in reward delivery, $\hat{R}_t(\tau)$ is zero for the first 2.5 s; after that it is set to the value of the reward,

$$\hat{R}_t(\tau) = \begin{cases} 0 & \tau < 2.5\,\text{s} \\ R & \tau \geq 2.5\,\text{s}. \end{cases} \tag{49}$$

Note that for the first 2.5 s of the trial, $-e_p$ carries a prediction of the upcoming reward from the olfactory input, $\mathbf{x}$. Once the reward is provided, the neuron represents the difference between the expected reward and the actual reward. That difference can be used to drive learning, via Hebbian plasticity

$$\bar{a}_j^t = \bar{a}_j^{t-1} + \eta_a e_p \bar{p}_j \tag{50}$$

where $\bar{a}_j$ is updated only after the reward has been presented. Importantly, $e_p$ is evaluated after the reward presentation.

Similarly, for the direct readout from $\mathbf{x}$ depicted in Fig. 7i, the reward is predicted by

$$\tau_r \frac{de_x}{d\tau} = -e_x + \hat{R}_t - \sum_i h_i x_i , \tag{51}$$

with $h_i$ again update via Hebbian plasticity,

$$h_i^t = h_i^{t-1} + \eta_h e_x x_i , \tag{52}$$

after the reward has been presented.

**Sparse coding.** The sparse coding model originally proposed by Olshausen and colleagues[10,52] can be applied to the model of olfactory learning as shown below. The basic idea is that the odor, denoted $\hat{\mathbf{c}}$, and the weight matrix, denoted $\hat{\mathbf{w}}$, that best explains the input, $\mathbf{x}$, should be close to the real $\mathbf{c}$ and $\mathbf{w}$. This means $\hat{\mathbf{c}}$ and $\hat{\mathbf{w}}$ can be estimated by performing stochastic gradient descent on the likelihood of the inputs, $\mathbf{x}$.

However, this is sub-optimal, primarily because uncertainty in $\hat{\mathbf{c}}$ and $\hat{\mathbf{w}}$ are ignored, even though they are important for data efficient learning[45]. In addition, for tractability, the prior over the odors is taken to be a continuous function, making it difficult to capture the fact that at any given time most odors are absent. These constraints make the learning algorithm inefficient.

The log likelihood of the data with respect to an unknown set of weights, denoted $\hat{\mathbf{w}}$, is given by

$$\log p(\mathbf{x}_t | \hat{\mathbf{w}}) = \log \left( \int p(\mathbf{x}_t | \mathbf{c}_t, \hat{\mathbf{w}}) p(\mathbf{c}_t) d\mathbf{c}_t \right) \tag{53}$$
$$\approx \log \left( p(\mathbf{x}_t | \hat{\mathbf{c}}_t, \hat{\mathbf{w}}) p(\hat{\mathbf{c}}_t) \right) + \text{const}.$$

In the second line, the integral was approximated with the maximum a posteriori

estimate $\hat{\mathbf{c}}_t = \arg \max_{\mathbf{c}} p(\mathbf{x}_t | \mathbf{c}, \hat{\mathbf{w}}) p(\mathbf{c})$. The objective function is thus given by

$$E_t \equiv \log p(\mathbf{x}_t | \hat{\mathbf{c}}_t, \hat{\mathbf{w}}) + \log p(\hat{\mathbf{c}}_t). \tag{54}$$

Because the noise on $\mathbf{x}_t$ is Gaussian (see Eq. (6)), the first term is a simple quadratic function. However, the second term, $\log p(\hat{\mathbf{c}}_t)$, requires further approximation to remove the delta function, and thus ensure differentiability of $E_t$ with respect to $\hat{c}_j$. To this end, we approximated the prior with a Gamma distribution: $p_c(\hat{c}_j) \propto \hat{c}_j^{k_c - 1} e^{-\hat{c}_j / \theta_c}$, for which the mean is $k_c \theta_c$. We used $k_c = 3$ and $\theta_c = c_o / 3$, ensuring a mean of $c_o$. Under this approximation, the objective function, $E_t$, becomes

$$E_t = \frac{-1}{2\sigma_x^2} \sum_i \left( x_i^t - \sum_j \hat{w}_{ij} \hat{c}_j^t \right)^2 + \sum_j \left( (k_c - 1)\log \hat{c}_j^t - \hat{c}_j^t / \theta_c \right). \tag{55}$$

We maximize the objective function via stochastic gradient descent, which occurs in two steps. In the first step, we maximize $E_t$ with respect to $\hat{\mathbf{c}}$,

$$\Delta \hat{c}_j \propto \frac{\partial E_t}{\partial \hat{c}_j} = \frac{1}{\sigma_x^2} \sum_i \hat{m}_i \hat{w}_{ij} + \frac{k_c - 1}{\hat{c}_j} - \frac{1}{\theta_c} , \tag{56}$$

where $\hat{m}_i$ is the analog of Eq. (17),

$$\hat{m}_i \equiv x_i - \sum_j \hat{w}_{ij} \hat{c}_j. \tag{57}$$

Once $\hat{c}_j$ has converged, we update the weights via

$$\Delta \hat{w}_{ij} \propto \frac{\partial E_t}{\partial \hat{w}_{ij}} = \frac{1}{\sigma_x^2} \hat{m}_i \hat{c}_j. \tag{58}$$

To prevent divergence of the weights, after each timestep we apply L-2 normalization (see Eq. (60b) below).

In summary, on each trial, $t$, first, the $\hat{c}_j$ ($j = 1, 2, ..., M$) are updated,

$$\hat{c}_j^t(\tau) = \hat{c}_j^t(\tau - 1) + \eta_c \left( \sum_i \hat{m}_i^t(\tau - 1)\hat{w}_{ij}^{t-1} + \sigma_x^2 \left[ \frac{2}{\hat{c}_j^t(\tau - 1)} - \frac{3}{c_o} \right] \right), \tag{59}$$

where the time step $\tau$ runs from 0 to 100,000 in each trial. At the end of trial $t$, the weights are then updated by

$$\tilde{w}_{ij} = \hat{w}_{ij}^{t-1} + \eta_w \hat{m}_i^t \hat{c}_j^t \tag{60a}$$

$$\hat{w}_{ij}^t = \frac{e}{c_o M} \frac{\tilde{w}_{ij}}{\sqrt{\sum_i \tilde{w}_{ij}^2 / N}} . \tag{60b}$$

The learning rates, $\eta_c$ and $\eta_w$, were manually tuned. We used $\eta_c = 0.00001$ and $\eta_w = 0.5$ unless stated otherwise.

**Simulation details.** The parameters used in the simulations are given in Table 1. Additional details of the simulations, from the *implementation of neural dynamics* to the setting of *Go/no go task*, are provided in Table 1.

**Implementation of neural dynamics.** The M/T cell activity, $m_i$, was defined relative to a baseline, denoted $m_{sp}$; in Fig. 3c, we plotted $\tilde{m}_i \equiv m_i + m_{sp}$. On each trial, $m_i$ was initialized to zero and $\bar{c}_j$ to $c_o$: $m_i(\tau = 0) = 0$ (i.e., $\tilde{m}_i(0) = m_{sp}$) and $\bar{c}_j(\tau = 0) = c_o$. In addition, the firing rates were lower-bounded by $m_i \geq -m_{sp}$ and $\bar{c}_j \geq 0$.

To avoid numerical instability, $\Psi(\alpha)$ in Eq. (21b) was approximated as

$$1/\Psi(\alpha) \approx \begin{cases} -\alpha & \frac{\alpha}{\sqrt{2}} < -10 \\ \frac{\exp(-\alpha^2/2)}{\sqrt{2\pi}\Phi(\alpha)} & -10 \leq \frac{\alpha}{\sqrt{2}} \leq 10 \\ 0 & 10 < \frac{\alpha}{\sqrt{2}} . \end{cases} \tag{61}$$

**Implementation of synaptic plasticity.** Both the feedforward and lateral weights were initially sampled from a log-normal distribution

$$w_{ij}^{t=0} = \log N(\mu_g^{init}, \sigma_g^{init}) , \tag{62}$$

with the variance and mean parameters set to

$$\sigma_g^{init} = 0.1 \tag{63a}$$

$$\mu_g^{init} = \frac{1}{2} \left( 1 - (\sigma_g^{init})^2 \right) - \log(c_o M) . \tag{63b}$$

The precision factors, $\rho_j$, were initialized as

$$\rho_j^{t=0} = \frac{c_o}{\sigma_x^2 Z_\rho} . \tag{64}$$

We used $Z_\rho = 0.5$, except in Fig. 6b and d, where we used $Z_\rho = 0.3$. The weights were lower-bounded by zero. As mentioned above, in the simulations we started $t$

**Table 1 Definitions and values of the parameters.**

| | Definition | Value |
|---|---|---|
| $M$ | The total number of odors presented and granule cell population | 100 (Figs. 3e, f, 4–6), 50 (Figs. 3c, d, 7) |
| $N$ | The total number of glomeruli | 400 (Figs. 3–6), 200 (Fig. 7) |
| $c_o$ | The probability of a odor being present | $3/M$, except Fig. 6b, d |
| $\sigma_x$ | The variance of noise on the glomeruli activity | 1.0 |
| $\sigma_\eta$ | The variance of the noise in the reward | 0.01 |
| $m_{sp}$ | The spontaneous firing rate of M/T cells | 5 Hz |
| $\tau_r$ | The timescale of firing rate dynamics | 50 ms |
| $T_{max}$ | The duration of each trials | 5000 ms, except for Fig. 3f |
| $t_{min}$ | The initial count for the global learning rate, $1/t$ | 100 [trials] |

from $t = t_{min}$ to suppress the influence of the initial samples. Recurrent inhibition, $J$, was initialized to $J_{jk} = 0.02 \times (1 - \delta_{jk})$.

**Learning with a fixed gain function.** In Fig. 5d, we fixed all $\lambda_j$ at 200 (gray) and 342 (black), while the $\rho_j^t$ were updated at each trial as in Eq. (35).

**Learning with a fixed learning rate.** Fixing the learning rate, $1/t\rho_j^t$, to a constant, denoted $\eta$, the learning rules for $w_{ji}^F$ and $w_{ij}^L$ are rewritten as

$$
\begin{aligned}
w_{ji}^{F,t} &= w_{ji}^{F,t-1} + \frac{\eta}{\sigma_x^2} \bar{c}_j \left[ m_i + \bar{c}_j w_{ji}^{F,t-1} \right] \\
w_{ij}^{L,t} &= w_{ij}^{L,t-1} + \frac{\eta}{\sigma_x^2} \bar{c}_j \left[ m_i + \bar{c}_j w_{ij}^{L,t-1} \right]
\end{aligned}
\tag{65}
$$

and $\lambda_j$ is given by

$$
\lambda_j = \frac{1}{\sigma_x^2} \left( \sum_{i=1}^{N} \left( w_{ji}^F \right)^2 + N\eta \right). \tag{66}
$$

**Go/no go task.** In the simulation of the go/no go task, we selected two odors ($j_+$ and $j_-$) out of $M$ total odors, then randomly presented one or the other with concentrations drawn from a Gamma distribution (as in Eq. (8), but $c_j > 0$ and $c_o = 1$). The reward associated with $j_+$ was $R = 1.0 + \zeta$ (i.e. $a_{j_+} = 1.0$), where $\zeta$ is the noise in the observed reward sampled from a zero-mean Gaussian with variance 0.01. The reward associated with $j_-$ was $R = \zeta$ (i.e. $a_{j_+} = 0.0$).

Learning of the circuit shown in Fig. 7h was done in two steps. First, the weights, $w_{ij}^F, w_{ij}^L$ and $w_{ij}^P$, and the precisions, $\rho_j$ and $\rho_j^P$, were learned with the unsupervised learning rules. During this unsupervised period, the reward, $R$, was kept at zero. After 4000 trials of unsupervised learning, we fixed $w_{ij}^F, w_{ij}^L, w_{ij}^P, \rho_j$, and $\rho_j^P$, then trained the weights $\bar{a}_j$ using Eq. (50).

The reward weights for the circuits in both Fig. 7h and i, $\bar{a}_j$ and $h_j$, respectively, were initialized to zero, and the learning rates were manually tuned to the largest stable rates ($\eta_a = 0.5$ and $\eta_h = 0.0015$). The latter learning rate was smaller because $\|\mathbf{x}\|$ is typically much larger than $\|\bar{\mathbf{p}}\|$, and also because the update of the $h_j$ was more susceptible to instability.

The classification performance was measured by the probability that the predicted and actual reward were both above 0.5 or both below 0.5,

$$
\text{performance} \equiv \langle \Theta[(R_t - 0.5)(-\hat{e}_p - 0.5)] \rangle, \tag{67}
$$

where $\hat{e}_p$ is the value of $e_p$ right before the reward delivery ($\hat{e}_p = e_p(\tau = 2.45\,\text{s})$). Note that, as mentioned above, $\hat{e}_p$ should converge to $-R_t$. Thus, the average error was defined to be

$$
\text{Average error} \equiv \left\langle (R_t + \hat{e}_p)^2 \right\rangle^{1/2}. \tag{68}
$$

**Performance evaluation.** In the following sections, we summarized the performance evaluation methods we employed in this study.

**Selectivity of granule cells.** Because the network is trained with an unsupervised learning rule, we cannot know which neuron encodes which odor. We thus estimated the selectivity of a neuron from the incoming synaptic weights using a bootstrap method. Specifically, on each trial, the odor $o(j)$ encoded by granule cell $j$ is determined by choosing the odor that yields the maximum covariance between the estimated weights, $\mathbf{w}^F$, and the true mixing weight, $\mathbf{w}$,

$$
o(j) = \arg\max_m \sum_{i=1}^{N} \left( w_{ji}^{F,t} - \langle w_{ji}^{F,t} \rangle_i \right)\left( w_{im} - \langle w_{im} \rangle_i \right). \tag{69}
$$

The selectivity can also be estimated from the activity of a neuron directly, by assuming that the granule cell with the highest activity to odor $j$ codes for odor $j$. Essentially the same result holds when we take this approach, although accurate

readout of selectivity requires a large number of trials. After learning, most neurons learn to encode one odor stably.

**Odor estimation performance.** Given the odor selectivity, $o(j)$, the original odors can be reconstructed by

$$
\hat{c}_j = \frac{\sum_{o(m)=j} \bar{c}_j}{\sum_{o(m)=j} 1}. \tag{70}
$$

The denominator is the number of neurons that encode odor $j$, which converges to one after successful learning. If both the denominator and the numerator were zero, we set $\hat{c}_j$ to 0. Performance was defined to be the correlation between the estimated odor concentration, $\hat{c}_j$, and the true concentration, $c_j$. Evaluation of performance on trial $t$ used $o(j)$ calculated from $w^{F,t-1}$, not from $w^{F,t}$. In Fig. 7e and f, we instead calculated the correlation between $\hat{p}_j$ and the true value of $\Theta[c_j]$ using the same method, where

$$
\hat{p}_j = \frac{\sum_{o_p(m)=j} \bar{p}_j}{\sum_{o_p(m)=j} 1}, \tag{71}
$$

using the piriform neuron selectivity $o_p(j)$.

**ROC curve.** We calculated the generalized ROC curves as in Fig. 7 of Grabska-Barwińska et al. (2017)[6] using $\hat{c}_j$. We first separated the trials based on the total number of odors presented, and then for each trial we calculated the number of true/false positives under various thresholds $\theta_{th}$. The true positive fraction is the fraction of presented odors above a threshold, $\theta_{th}$, whereas the false positive count is the number of absent odors above a threshold, $\theta_{th}$. The threshold, $\theta_{th}$, was varied from $10^{-6}$ to $10^1$ in a log scale, with an ~20% increase on every step.

**Weight error.** Given $o(j)$, the error between the learned feedforward weight, $w_{ij}^F$, and the true mixing weight, $w_{ij}$, was calculated by

$$
d_w^{F,t} \equiv \frac{1}{M} \sum_j^M \sqrt{\frac{1}{N} \sum_i^N \left( w_{ji}^{F,t}/Z_{j,t}^w - w_{i,o(j)} \right)^2}, \tag{72}
$$

where $Z_{j,t}^w = \sum_i w_{ji}^{F,t}/\sum_i w_{i,o(j)}$. For ease of comparison, in Fig. 6b the weight errors were scaled by 7/3, so that the initial error was similar to the errors shown in Fig. 6a.

**Lifetime sparseness.** For the measurement of the lifetime sparseness[19], we first presented individual odors $m = 1, 2, \ldots, M$, then recorded the activity of granule cells $\{\bar{c}_j^{(m)}\}$. Subsequently, we calculated the sparseness using

$$
S_j \equiv \frac{\left( \frac{1}{M} \sum_{m=1}^M \bar{c}_j^{(m)} \right)^2}{\frac{1}{M} \sum_{m=1}^M \left( \bar{c}_j^{(m)} \right)^2}. \tag{73}
$$

The lifetime sparseness, $S_j$, takes a small value ($S_j \simeq 0$) if the activity is sparse, while $S_j \lesssim 1$ is satisfied if the activity is uniform/homogeneous. Because of this, the lifetime sparseness is sometimes defined as $\tilde{S}_j \equiv 1 - S_j$[53].

**Reporting summary.** Further information on research design is available in the Nature Research Reporting Summary linked to this article.

## Data availability

The main source codes of the simulations and the data analysis, from which our simulation date was generated, are publicly available at https://github.com/nhiratani/olfactory_learning. The rest are available from the corresponding author.

## Code availability

The main codes for simulations and data analysis are publicly available as mentioned above, at http://github.com/nhiratani/olfactory_learning.

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

## Acknowledgements

This work was supported by the Gatsby Charitable Foundation and the Wellcome Trust (110114/Z/15/Z).

## Author contributions

NH and PEL designed the research; NH performed the research; NH analyzed the data; and NH and PEL wrote the paper.

## Competing interests

The authors declare no competing interests.
