## [Peer Review File · Nature Communications]

Reviewers' Comments:

Reviewer #1:

Remarks to the Author:

Hiratani and Latham present a Bayesian framework for learning in the olfactory bulb. I feel unqualified to comment on the computational aspects of the work, but as an experimentalist I see a lot to like. The paper makes strong, testable predictions of how odor responses should evolve during latent and reward-based learning. I predict that this paper will have great value for both theorists and experimentalists.

I do have some issues with the paper from an ethological perspective:

L106-2009 "This is at odds with the demands of a realistic environment, where the existence of an odor is almost

always far more important than its concentration"

Nothing could be further from the ethological truth! Detecting and identifying an odor is useless without being able to locate its source, and most animals use olfaction to find things (Baker, ... Nagel, and Smear 2019). Therefore, as in other sensory modalities, olfaction faces both "what" and "where" problems. "Where" can only be judged by spatiotemporal patterns of odor concentration. All of which is to say that estimating odor concentration is not a "drawback" of the model.

Another biological truth of the olfactory system is that it is fast (eg, Shusterman et al 2011) and can enable very short reaction times (eg., Uchida & Mainen 2003, Resulaj & Rinberg, 2014), and especially needs to be so when an animal navigates an odor plume using odor concentration cues. In this connection, fig 3C&D shows that odor responses diverge more rapidly after learning. I would like to get a better intuition for why the model shows this behavior. Furthermore, I wonder how the learning curves of figure 4 will look under realistic timing constraints (ie, one sniff decisions).

Lastly, I question the "spike-and-slab" prior distribution, specifically the slab part. We know very little about natural olfactory scene statistics, but I'm not sure the shape of the slab is realistic. Are non-zero odor concentrations most often very low? I don't think so. An animal quite often experiences intense odors, and must retain dynamic range to sense concentration changes in this regime. How sensitive are the conclusions of the modelling to the assumed shape of the slab? Does a model with a flatter slab still learn and discriminate as well?

-Matt Smear

Reviewer #2:

Remarks to the Author:

Hiratani and Latham present a paper entitled "Rapid bayesian learning in the mammalian olfactory system". The proposed ideas seem interesting from a mathematical point of view, but a lack of clarity makes it difficult to evaluate how this applies to the olfactory system or sensory systems in general. In the end a neural model is constructed that learns (what?) and seems disconnected from the math show in the beginning.

- It is not clarified what is being learned? At first one may think its the "weights" between odors and sensory neurons (this should really be referred to as affinities), because these are referred to as :the system needs to learn what they are", but then it seems that weights between mitral and granule cells

are being learned

- what is the goal? The performance measure seems to indicate the goal is to reconstruct the inputs, but where and why?
- the authors talk about odor learning but later explain that the model learns to detect the concentration of an odor ?Which is it ? Identity or concentration?
- How is this "rapid" learning evaluated? What is the goal of the learning? It is to create different representations for different odorants? This is not at all clear. One can see that MC outputs change as a function of learning but nowhere is it stated what the expectations are here
- Is the system learning odor concentrations, odor identities? Or just learning to differentiate? What are the limits of these differentiation? How similar can odors be?
- It seems impossible to learn an odorant in an unsupervised manner. What does that mean? If the system is to learn the identity of an odorant, how would it know this in an unsupervised manner?
- It seems that maybe the authors are going after separation of odors, which would be plausible from the figures, but they are not testing the limits of their approach or showing how well it works for more odors, more overlapping odors ..
- do all odors have to be learned at the same time?
- the assumption that odors are sparse is crucial to the approach yet seems far fetched. The mammalian olfactory system is constantly inhaling odorized air and odors are every where. How well can this work without this assumption?
- the comparison the sparse coding is only useful if one is told what is "good" . The performance measures says to reconstruct the input, but this is definitely not what the olfactory system wants to do ..
- it is not clear how the theoretical statements in the beginning aid the model. why not just make a model since it then includes hebbian plasticity and so on? While the neural model makes some interesting discoveries it is not clear how the bayesian theory contributes to the model in the end.

Overall there might be an interesting model of olfactory learning here, but to many details as to what is being learned and how have been omitted. The manuscript is very unclearly written and jumps from topic to topic. Assumptions are not clearly explained and the goals are not clarified.

Reviewer #3:

Remarks to the Author:

The paper by Hiratani and Latham outlines a clever piece of theory which uses Bayesian inference arguments and assumptions of a sparse stimulus world to derive physiology and circuitry structure in olfactory processing. The work is well done and introduces some novel concepts that can easily extend beyond olfaction to more general neuronal coding. Overall, I enjoyed the paper. I do have some queries/suggestions that I hope improve the manuscript.

Major

1. The authors show that the transfer nonlinearity F has features consistent with neuronal recording when the prior is sparse, or at least biased to very weak concentrations. This is a nice result and the underlying mathematics in deriving Eqs 31 and 32 is elegant. However, the authors also derive a nonlinearity G for the second moment of the concentration in Eq 33. Is there anything to be gained in understanding how the variance of c_j depends on the prior? Basically, is the analogue of Fig 5A but for G (or $G - F^2$) at all interesting or predictive?

2. Fig 5A shows how both the 'spike-and-slab' prior and the exponential prior give different cellular transfer functions F . The text surrounding line 142ish argues that both are more consistent with

physiology than the non-negative prior. This is certainly an interesting point, and arguably one of the triumphs of the study. The fact that sparse stimulus distributions predict physiology is indeed quite cool. However, the paper seems to favor the spike-and-slab prior over the exponential prior. Why? What is the evidence for the spike-and-slab other than it has a non-infinitesimal mass at zero concentration? Indeed, the transfer function F for the exponential prior seems more in line with the in vivo whole cell recordings when mean spike rate is plotted against mean V_m (a la D. Ferster).

3. The authors should discuss why that in Fig. 6A the fixed fast rate learning gives better performance early on than the fixed slow rate learning, yet for large trial numbers the order is switching. I assume that slow rate learning remembers more past experience and thus is the better performer after sufficient experience is given.

4. While the main text does introduce Bayesian inference reasonably well, any intuition for the variational approximation outlined in sections 4.1.1 and 4.1.2 is absent. I understand the demands of streamlining the work for a general audience, but the variational approximation is central to the work. It is a shame that no intuition to this work can be given to a general reader.

Minor

1. The paragraph beginning on line 100 discusses the optimality of algorithm, in particular how the approximation via q affects optimality. Indeed, the Bayesian algorithm compares well against the sparse coding model. However the paragraph focuses primarily on the speed of learning rather than the overall performance. Fast, efficient learning is all well and good, but this does not really test the optimality as the first sentence of the paragraph suggests. I understand that a brute force calculation of optimal performance is not feasible. Then a simple rewrite of the paragraph so that it is clear how the sparse and the variational Bayesian algorithms are actually compared to one another is warranted.

2. Maybe I missed this, but does the dendro-dendritic synaptic connections between granule and mitral cells for a certain symmetry between w^L and w^F ? Basically, if a mitral cell couples to a granule cell then the reverse must also be true. The Hebbian plasticity rule in 5a and 5b also suggests that the reciprocal connections are either both strong or are both weak. Is there any experimental evidence for this?

Please find the revised manuscript attached. Major changes are shown in red in the manuscript. Some new figures were added as supplementary figures (Supplementary Figures 1 and 2) due to the tight page constraint. In addition to the revisions in response to the comments from the reviewers, we have made the following minor changes,

1. We added titles for all the figures in accordance with the format guideline.
2. We removed subheadings from the Discussion.
3. We added data and code availability statements.

Reviewer #1 (Remarks to the Author):

Hiratani and Latham present a Bayesian framework for learning in the olfactory bulb. I feel unqualified to comment on the computational aspects of the work, but as an experimentalist I see a lot to like. The paper makes strong, testable predictions of how odor responses should evolve during latent and reward-based learning. I predict that this paper will have great value for both theorists and experimentalists.

I do have some issues with the paper from an ethological perspective:

Comment #1.1

L196-197 "This is at odds with the demands of a realistic environment, where the existence of an odor is almost always far more important than its concentration" Nothing could be further from the ethological truth! Detecting and identifying an odor is useless without being able to locate its source, and most animals use olfaction to find things (Baker, ... Nagel, and Smear 2019). Therefore, as in other sensory modalities, olfaction faces both "what" and "where" problems. "Where" can only be judged by spatiotemporal patterns of odor concentration. All of which is to say that estimating odor concentration is not a "drawback" of the model.

Reply #1.1

Thank you for pointing that out. We agree that the statement was too strong. What we intended to claim was that the concentration is usually not a reliable indicator of the valence of the odor. It is therefore, useful to have a concentration-invariant representation of the presented odors, as suggested from the indication of concentration-invariance in piriform cortex. However, of course that does not deny the importance of concentration information in foraging. Please also note that the model developed in Figs. 2-6 can indeed recover the odor concentration. We now acknowledge this in the first paragraph of the subsection "Learning a concentration invariant representation and valence" based on your comment (L226-228).

Comment #1.2

Another biological truth of the olfactory system is that it is fast (eg, Shusterman et al 2011) and can enable very short reaction times (eg., Uchida & Mainen 2003, Resulaj & Rinberg, 2014), and especially needs to be so when an animal navigates an odor plume using odor concentration cues. In this connection, fig 3C&D shows that odor responses diverge more rapidly after learning. I would like to get a better intuition for why the model shows this behavior. Furthermore, I wonder how the learning curves of figure 4 will look under realistic timing constraints (ie, one sniff decisions).

Reply #1.2

First, about the acceleration of the inference, at the beginning of the learning the synaptic weights are relatively

homogeneous. As a result, all granule cells receive similar amplitude of feedforward inputs, causing competition through lateral inhibition, which eventually enables the circuit to detect the odors. In contrast, after learning, a small numbers of granule cells start to receive strong feedforward input. Because of this, there is less competition among the cells, which makes detection faster. Note that because both feedforward and feedback synaptic weights become sparse after learning, the exact mechanism of the acceleration is expected to be somewhat complicated.

Regarding the stimulation duration, to better characterize the effect of the time constant, we performed a series of simulation with different stimulation duration (Fig. 3F). We found that even if we chose a short presentation, the circuit still learned to detect odors. In particular, the typical time of one sniff cycle (200-500 ms) was enough for successful learning. The model failed to learn the odors only when the stimulation duration became comparable to the time constant of the firing rate dynamics (50 ms). It might be possible to overcome this limitation by considering a spike-based implementation, but we will leave that for future work.

We've added a brief explanation of this point (L119-121).

Comment #1.3

Lastly, I question the "spike-and-slab" prior distribution, specifically the slab part. We know very little about natural olfactory scene statistics, but I'm not sure the shape of the slab is realistic. Are non-zero odor concentrations most often very low? I don't think so. An animal quite often experiences intense odors, and must retain dynamic range to sense concentration changes in this regime. How sensitive are the conclusions of the modelling to the assumed shape of the slab? Does a model with a flatter slab still learn and discriminate as well?

Reply #1.3

Let us first note that the concentration variable c represents the log-concentration in the model. If we instead consider linear concentration by making the replacement $c \rightarrow \log c_{\text{linear}}$, we get power law tails in c_{linear} (Supplementary Figure 1A, B). In that respect, the distribution of the concentration is already large. To test sensitivity to the shape of the slab, we used a wider concentration distribution for the slab: e^{-c} rather than $c^2 e^{-3c}$ (purple versus orange lines in Supplementary Figure 1A, B). Learning performance was very similar even under a broad concentration distribution (Supplementary Figure 1C). We thank the reviewer for this comment, as it is important; we now address this point in the main text (L121-L123).

Reviewer #2 (Remarks to the Author):

Hiratani and Latham present a paper entitled "Rapid bayesian learning in the mammalian olfactory system". The proposed ideas seem interesting from a mathematical point of view, but a lack of clarity makes it difficult to evaluate how this applies to the olfactory system or sensory systems in general. In the end a neural model is constructed that learns (what?) and seems disconnected from the math show in the beginning.

General Reply

Judging from the reviewer's comments, we believe that the main issue is with what is being learned. We now include a paragraph in the introduction that, hopefully, will make this point clearer (L7-15). In brief, we are considering unsupervised learning: animals have access only to glomeruli activity, and have to infer which patterns of activity correspond to which odors. This is essentially a clustering problem. After learning, the animal will be able to recognize a particular pattern of activity as corresponding to an odor (or set of odors) it has experience before.

In our model, each odor activates a different granule cell. So, for instance, every time odor A is present, granule cell A is active; and every time odor B is present, granule cell B is active. Thus, although the animal cannot "name" the identity of odors, its granule cell activity effectively tells it about the odors that are present. Here we mainly focused on the learning of odor representation by the granule cells, but we find that odors can also be read out from piriform cortex (Fig. 7A-D).

To detect odors, the system has to learn the OSN affinities (which we refer to as weights) for the odors, and store the affinities as synaptic weights. In our model, the weights are stored as the connection strengths of dendro-dendritic connections between mitral/tufted cells and granule cells. In addition we investigate the ability of our model to form odor-reward associations. We find that after unsupervised learning, these associations are formed rapidly.

We hope that provides a clearer picture of the goal of our study. We have augmented this below with replies to specific comments.

Comment #2.1

- It is not clarified what is being learned? At first one may think its the "weights" between odors and sensory neurons (this should really be referred to as affinities), because these are referred to as :the system needs to learn what they are", but then it seems that weights between mitral and granule cells are being learned

Reply #2.1

Hopefully the General Reply above addressed this point: the affinities are indeed being learned, and they are stored by the efficacy of the synaptic connections between the mitral/tufted cells and granule cells.

Comment #2.2

- what is the goal? The performance measure seems to indicate the goal is to reconstruct the inputs, but where and why?

Reply #2.2

As mentioned in the General Reply, the first goal of the animal is to develop a unique mapping from odors to granule cell activity. After that, the goal is to learn odor-reward associations. In principle, odor-reward association can be learned directly from OSN activity (Fig. 7E(ii)). However, learning tends to be slow and imprecise in that case (purple lines in Fig. 7F), because neural activity of OSNs is dense and noisy (Fig. 1 right). On the other hand, if the odor-reward association is learned from the reconstructed odors (Fig. 7E(i)), learning is fast and accurate (magenta lines in Fig. 7F). This is mainly why we focused on the learning of the input odor reconstruction. We now include a brief explanation of this point in the introduction (L7-15), and we mention it again on L50-52.

Comment #2.3

- the authors talk about odor learning but later explain that the model learns to detect the concentration of an odor ?Which is it ? Identity or concentration?

Reply #2.3

Both identity and concentration are inferred in the model. The model infers the concentration of the odor as shown in Fig. 3C (here, the horizontal dotted lines in the bottom panels represent the true concentrations), and also the probability that an odor is present (Fig. 7B-D).

Comment #2.4

- How is this "rapid" learning evaluated? What is the goal of the learning? It is to create different representations for

different odorants? This is not at all clear. One can see that MC outputs change as a function of learning but nowhere is it stated what the expectations are here

Reply #2.4

The speed of learning is evaluated via performance-versus-time curves (Fig. 4A). As stated in the General Reply (and, now, in the introduction), the goal is indeed to create different representations for different odorants.

To measure performance (which is not trivial in an unsupervised setting), we first estimated the selectivity of each granule cell, and then recovered the odor concentrations based on the activity of those neurons. The performance was then given by the correlation between the recovered odors and the true odors (“Performance evaluation” in Methods). We also measured the performance by the similarity between the learned synaptic weights and the true affinities between the odors and OSNs (weight error). These two measures were consistent with each other. We now include a brief explanation on the evaluation method (L116-119).

Comment #2.5

- Is the system learning odor concentrations, odor identities? Or just learning to differentiate? What are the limits of these differentiation? How similar can odors be?

Reply #2.5

The answer to the first question is “both” (see Reply #2.3). In an unsupervised setting, this learning of odor concentration and identity is done by learning different representations for different odorants.

We did not directly check performance versus similarity of odors, but we investigated the size of a mixture that could be recovered. The ROC curves (Fig. 3E) show that the model can reliably recover the presented odors when 1-3 odors are presented (purple and blue lines), but the performance gets slightly worse as the number of simultaneously presented odors increases. This result implies that the model would struggle to differentiate two mixtures composed of more than seven odors with only one different component.

Comment #2.6

- It seems impossible to learn an odorant in an unsupervised manner. What does that mean? If the system is to learn the identity of an odorant, how would it know this in an unsupervised manner?

Reply #2.6

Indeed, the brain cannot tell which input pattern corresponds to which odor identity, but the brain can still learn to recover the “features” corresponding to individual odors. Please refer to General Reply above.

Comment #2.7

- It seems that maybe the authors are going after separation of odors, which would be plausible from the figures, but they are not testing the limits of their approach or showing how well it works for more odors, more overlapping odors ..

Reply #2.7

Though indeed we did not consider different degrees of odor overlap, we now tested the number of odors in the mixture that could be recovered, as we discussed in Reply #2.5 (and see Fig. 3E).

Comment #2.8

- do all odors have to be learned at the same time?

Reply #2.8

Our preliminary simulation results (not shown) suggest that odors do not have to be learned at the same time.

However, in the model, we assumed, for simplicity, that all odors were learned at the same time.

Comment #2.9

- the assumption that odors are sparse is crucial to the approach yet seems far fetched. The mammalian olfactory system is constantly inhaling odorized air and odors are every where. How well can this work without this assumption?

Reply #2.9

We agree that animals are constantly inhaling odorized air, and odors are every where. However, what we mean by sparseness is that at any one time, only a small number of odors are present at a detectible concentration. And indeed, if a large number of odors were present simultaneously, the model would struggle to detect them. However, animals too tend to struggle under such a condition, unless intense reward-based learning is conducted (e.g., Chapuis & Wilson, *Nat Neurosci*, 2010; Jinks. & Laing, *Perception*, 1999).

Comment #2.10

- the comparison the sparse coding is only useful if one is told what is "good" . The performance measures says to reconstruct the input, but this is definitely not what the olfactory system wants to do ..

Reply #2.10

Reconstruction of odors is indeed not necessarily the primary goal of the olfactory system. However, it is critical for making rapid odor-reward associations (L256-272). Thus, we believe that the reconstruction of the presented odors is an important sub-goal of the olfactory system.

Comment #2.11

- it is not clear how the theoretical statements in the beginning aid the model. why not just make a model since it then includes hebbian plasticity and so on? While the neural model makes some interesting discoveries it is not clear how the bayesian theory contributes to the model in the end.

Reply #2.11

We assume that by "just make a model", the reviewer means an approach in which a neural circuit model is constructed based on the known anatomy, and it's combined with a Hebbian type learning rule to updates the weights. Such a bottom-up approach is indeed useful in many occasions, but a top-down approach based on Bayesian statistics is also essential for understanding the brain.

Indeed, our model based on Bayesian statistics lead us to non-trivial predictions which wouldn't be obtained otherwise. For instance, we found that the spike-and-slab prior yielded a complicated but biologically realistic transfer function. Moreover, our investigation of the neural substrate of uncertainty representation provided a prediction of the relationship between the lifetime sparseness and the malleability of a neuron.

In addition, please note that most of the approximations were done with the variational Bayesian method, which enabled us to introduce some biological constraints into the model while using the Bayesian framework. We thus believe that Bayesian statistics is a necessary basis of our work.

Comment #2.12

Overall there might be an interesting model of olfactory learning here, but too many details as to what is being learned and how have been omitted. The manuscript is very unclearly written and jumps from topic to topic. Assumptions are not clearly explained and the goals are not clarified.

Reply #2.12

We hope our replies clarified the objective and the results of our work.

Reviewer #3 (Remarks to the Author):

The paper by Hiratani and Latham outlines a clever piece of theory which uses Bayesian inference arguments and assumptions of a sparse stimulus world to derive physiology and circuitry structure in olfactory processing. The work is well done and introduces some novel concepts that can easily extend beyond olfaction to more general neuronal coding. Overall, I enjoyed the paper. I do have some queries/suggestions that I hope improve the manuscript.

Major

Major comment #3.1

1. The authors show that the transfer nonlinearity F has features consistent with neuronal recording when the prior is sparse, or at least biased to very weak concentrations. This is a nice result and the underlying mathematics in deriving Eqs 31 and 32 is elegant. However, the authors also derive a nonlinearity G for the second moment of the concentration in Eq 33. Is there anything to be gained in understanding how the variance of c_j depends on the prior? Basically, is the analogue of Fig 5A but for G (or $G - F^2$) at all interesting or predictive?

Reply #3.1

Thank you for a positive comment on our analysis. Our theory predicts that the change in the weight precision, ρ_j , should depend on the second moment, $\langle c_j^2 \rangle$ (Eq. 27a and 33), not the variance, $\langle (c_j - \langle c_j \rangle)^2 \rangle$. Because of this, the shape of G resembles F^2 , and $G - F^2$ takes near zero values regardless of input current (Supplementary Figure 2A and 2B). Thus, we expect that the prior dependence of the variance of c_j has very little effect on the learning dynamics. We have added a short explanation of this in the main text (L172-L174).

Major comment #3.2

2. Fig 5A shows how both the 'spike-and-slab' prior and the exponential prior give different cellar transfer functions F . The text surrounding line 142ish argues that both are more consistent with physiology than the non-negative prior. This is certainly an interesting point, and arguably one of the triumphs of the study. The fact that sparse stimulus distributions predict physiology is indeed quite cool. However, the paper seems to favor the spike-and-slab prior over the exponential prior. Why? What is the evidence for the spike-and-slab other than it has a non-infinitesimal mass at zero concentration? Indeed, the transfer function F for the exponential prior seems more in line with the in vivo whole cell recordings when mean spike rate is plotted against mean V_m (a la D. Ferster).

Reply #3.2

The problem with a pure exponential prior is that, it would imply that we would be constantly bombarded with a large number of odors, making inference almost impossible. In fact, it is known that humans, at least, cannot infer more than three odors at a time (Jinks, A. & Laing, D.G. A limit in the processing of components in odor mixtures. *Perception* 28, 395–404, 1999), and even with three odors they do poorly. Behavioral experiments in rodents also suggest that they struggle to differentiate two mixtures composed of many similar odorants (eg. Chapuis & Wilson, *Nat Neurosci*, 2010). Thus, the "spike" in our spike-and-slab model is critical, since it's what makes the odors sparse (on average about three are present in our model; see Sec. 2.4). The shape of the slab, however, is not so important. In the main text, we used $c^2 e^{-3c}$. However, we could change it to an exponential, e^{-c} , with very little change in performance (Supplementary Figure 1).

Still, that does not exclude the suitability of the exponential prior for other sensory systems. In particular, one corollary of our analysis is that, depending on the corresponding prior, the shape of transfer function of a neuron should be different. For instance, in the early visual system, the prior on visual features rather resembles an exponential distribution (Olshausen & Field, 1997), instead of the spike-and-slab prior. This might explain why the transfer function of neuron recorded from the visual cortex, such as the curve in Anderson, Carandini, and Ferster (2000), resembles the curve derived from the exponential prior, as the reviewer pointed out. Having said that, verification of this hypothesis requires a well controlled comparison of the transfer functions. We briefly mentioned this speculation in the previous manuscript, but we have now extended it slightly (L169-L172).

Major comment #3.3

3. The authors should discuss why that in Fig. 6A the fixed fast rate learning gives better performance early on than the fixed slow rate learning, yet for large trial numbers the order is switching. I assume that slow rate learning remembers more past experience and thus is the better performer after sufficient experience is given.

Reply #3.3

From the Bayesian perspective, a fixed slow-rate learning is suboptimal, because it overweighs the prior on the synaptic weight, while neglecting the likelihood provided by a new sample. On the other hand, under a fixed fast-rate learning, the model under-reacts to the prior, while relying too much on the likelihood, again resulting a sub-optimal update. Because the optimal learning rate decreases as the model experiences more trials, the fixed fast-rate learning is initially closer to the optimal but slowly becomes suboptimal. By contrast, the fixed slow-rate learning tends to perform better at a late phase. This is essentially why you see switch in the performance of the two learnings, though in reality that is not always the case due to the trajectory dependence of learning. We now mention this in the text (L210-213).

Given that the prior on the weight is formed from past experiences, the fixed slow-rate learning indeed values the collective past experiences more at each update, as the reviewer suggested. Still, let us note that in our problem setting, the past experience affects the update only as a collective prior, not as memories.

Major comment #3.4

4. While the main text does introduce Bayesian inference reasonably well, any intuition for the the variational approximation outlined in sections 4.1.1 and 4.1.2 is absent. I understand the demands of streamlining the work for a general audience, but the variational approximation is central to the work. It is a shame that no intuition to this work can be given to a general reader.

Reply #3.4

We have now included a graph explains the variational approximation (Fig. 2C). The left panel of the figure describes the true posterior of two odors, while the right panel is its variational approximation with a factorized distribution. We believe this graph characterizes how a variational approximation with the factorized distribution captures the mean and variance of each odor, while ignoring the structure of covariance. We included a description of the figure in the main text (L71-L74).

Minor Comment #3.1

1. The paragraph beginning on line 100 discuss the optimality of algorithm, in particular how the approximation via q affects optimality. Indeed, the Bayesian algorithm compares well against the sparse coding model. However the paragraph

focuses primarily on the speed of learning rather than the overall performance. Fast, efficient learning is all well and good, but this does not really test the optimality as the first sentence of the paragraph suggest. I understand that a brute force calculation of optimal performance is not feasible. Then a simple rewrite of the paragraph so that it is clear how the sparse and the variational Bayesian algorithms are actually compared to one another is warranted.

Reply #3.1

We have now rephrased the corresponding paragraph to make it clear which comparison was actually made (L124-127). Please note that, this paragraph does explain how we compare the Bayesian and sparse coding approaches: both in speed of learning (Fig. 4A) and asymptotic performance (Fig. 4C).

Minor Comment #3.2

2. Maybe I missed this, but does the dendro-dendritic synaptic connections between granule and mitral cells for a certain symmetry between w^L and w^F ? Basically, if a mitral cell couples to a granule cell then the reverse must also be true. The Hebbian plasticity rule in 5a and 5b also suggests that the reciprocal connections are either both strong or are both weak. Is there any experimental evidence for this?

Reply #3.2

Indeed, the weights w^L and w^F eventually converge to similar values after a sufficient learning, though we initialized these values independently. To the best of our knowledge, there is no directly evidence for this symmetry. It is known that depolarization of the spine triggers GABA release, as it causes Ca^{2+} influx through voltage-dependent calcium channel (Halabisky, et al., *J Neurosci* 2000). This mechanism implies that if the mitral cell spike causes large depolarization, a proportionally large reciprocal GABA release is expected. However, this is not the only mechanism for the GABA release onto the mitral cell dendrite (Egger et al., *J Neurosci* 2005), and it is still unclear how overall interaction through the dendro-dendritic connection is regulated.

Reviewers' Comments:

Reviewer #1:

Remarks to the Author:

The authors have addressed comments to my satisfaction.

Sincerely,

Matt Smear

Reviewer #2:

Remarks to the Author:

The authors have included more clarity as to what is learned and why and the paper is easier to read. I do have one request which is to not refer to receptor affinities as weights because this is confusing, why not use the correct term? While I still think this is a bit remote from the biological system, this is a fascinating paper with a number of interesting ideas which will certainly change my thinking about the first stage of olfactory learning.

Reviewer #3:

Remarks to the Author:

The authors did a solid job at answering my concerns. I am happy to support publication.

Reply to Reviewers

Reviewer #2 (Remarks to the Author): *The authors have included more clarity as to what is learned and why and the paper is easier to read. I do have one request which is to not refer to receptor affinities as weights because this is confusing, why not use the correct term? While I still think this is a bit remote from the biological system, this is a fascinating paper with a number of interesting ideas which will certainly change my thinking about the first stage of olfactory learning.*

Reply: Thank you for insightful comments on the manuscript. We have now made more clear that, in the generative model, w_{ij} represents affinities, at L7, L10, L39, and in the line above Eq. (7a). We retained the variable name w , as it is the mixing weights in signal processing terminology, and it is later represented by synaptic weights in the inference model.